# S100 Proteins as Novel Therapeutic Targets in Psoriasis and Other Autoimmune Diseases

**DOI:** 10.3390/molecules27196640

**Published:** 2022-10-06

**Authors:** Katarzyna Kurpet, Grażyna Chwatko

**Affiliations:** 1Doctoral School of Exact and Natural Sciences, University of Lodz, Banacha 12/16, 90-237 Lodz, Poland; 2Department of Environmental Chemistry, Faculty of Chemistry, University of Lodz, Pomorska 163/165, 90-236 Lodz, Poland

**Keywords:** psoriasis, S100 proteins, koebnerisin, antimicrobial peptides and proteins, immune-mediated inflammatory diseases, biomarkers

## Abstract

Psoriasis is one of the most common inflammatory skin diseases affecting about 1–3% of the population. One of the characteristic abnormalities in psoriasis is the excessive production of antimicrobial peptides and proteins, which play an essential role in the pathogenesis of the disease. Antimicrobial peptides and proteins can be expressed differently in normal and diseased skin, reflecting their usefulness as diagnostic biomarkers. Moreover, due to their very important functions in innate immunity, members of host defense peptides and proteins are currently considered to be promising new therapeutic targets for many inflammatory diseases. Koebnerisin (S100A15) belongs to an S100 family of antimicrobial proteins, which constitute the multigenetic group of calcium-binding proteins involved in ion-dependent cellular functions and regulation of immune mechanisms. S100A15 was first discovered to be overexpressed in ‘koebnerized’ psoriatic skin, indicating its involvement in the disease phenotype and the same promising potential as a new therapeutic target. This review describes the involvement of antimicrobial peptides and proteins in inflammatory diseases’ development and therapy. The discussion focuses on S100 proteins, especially koebnerisin, which may be involved in the underlying mechanism of the Köebner phenomenon in psoriasis, as well as other immune-mediated inflammatory diseases described in the last decade.

## 1. Introduction

Psoriasis is a chronic, autoimmune inflammatory disease with dysregulation of antimicrobial peptides and proteins (AMPs) in the immune system [1,2,3]. AMPs have a direct and broad spectrum of pathogen killing, anti-biofilm activity, as well as strong immunomodulatory effects [4,5,6,7]. These remarkable properties of host defense peptides and proteins have recently captured researchers’ attention to the possible use of AMPs as innovative drug candidates for the treatment of infectious diseases caused by multidrug-resistant microorganisms and as novel immunomodulatory therapies [8,9,10]. AMPs include members of the largest multigenic Ca^2+^-binding EF-hand motif S100 family that gained great interest as key players in the pathogenesis of inflammatory diseases such as psoriasis [2,11,12,13,14,15]. S100 proteins are expressed only in vertebrates in a cell- and tissue-dependent manner. These biomolecules developed many specific intracellular functions such as the regulation of proliferation, calcium homeostasis, cell growth and differentiation, enzyme activities, and more. The extracellular activities of the S100 proteins appear to be specific for selected members of this family and include chemotaxis, involvement in host innate and adaptive immune responses, tissue development and repair, as well as leukocyte infiltration. Due to the wide range of functions of the S100 proteins, various diseases such as chronic inflammatory diseases are associated with altered levels of particular S100 proteins, which can be induced under pathological conditions in a cell type that does not express it under normal physiological conditions. For this reason, S100 proteins are being discussed as potential biomarkers for many diseases as well as new therapeutic targets [1,12,13,15,16,17,18]. Koebnerisin (S100A15) belongs to the S100 protein family and was first discovered to be overexpressed in ‘koebnerized’ psoriatic skin, indicating its involvement in the disease phenotype [19,20,21]. In the model of psoriasis, inflammation of keratinocytes is mediated by upregulation of S100A15. Koebnerisin drives the production of some proinflammatory cytokines, attracts immunocytes, and further potentiates the inflammatory cascade [22]. The pro-inflammatory environment then stimulates overexpression of AMPs, creating a psoriatic feedback loop [23]. Thus, koebnerisin appears to be an excellent candidate for a biomarker in psoriasis. In this review, we aim to provide a short overview of the pathogenesis of psoriasis and the role of antimicrobial peptides and proteins in the biological systems far beyond their anti-microbial activity. The possible applications of AMPs as a new class of therapeutic agents in some diseases as well as major difficulties associated with their biological activity for clinical application are discussed. In addition, the function of S100 protein family in human diseases, with particular emphasis on the role of koebnerisin in the early stages of the inflammatory cascade in psoriasis, are addressed.

## 2. Pathogenesis of Psoriasis

Psoriasis is defined as a systemic, chronic immune-mediated inflammatory disease that affects approximately 1–3% of the population worldwide [24]. It is a non-contagious, papule-exfoliative disease, affecting women and men equally [25,26,27]. Psoriasis can occur at any age, but it usually appears for the first time between 15 and 25 years of age (type I psoriasis) and after 40 years (type II psoriasis). White people suffer from psoriasis more often than black people. This disease is characterized by alternating severity and remission of disease symptoms, which include the formation of skin lesions of varying severity in the form of symmetrical, sharply delimited erythematous papules, usually covered with white, shiny scales [28,29]. Skin lesions in psoriasis can occupy symmetrically all areas of the body, but the most characteristic places of lesions are elbows and knees, straight surfaces of the upper and lower limbs, lumbosacral region, trunk, hands and feet, hairy scalp, skin folds, nails, and under the breasts and genitals [26,27,30]. When the scales are scraped off the surface of the lump, the scales fall off, forming thin flakes reminiscent of candle scrapings (a sign of a stearic candle) [25]. Punctate bleeding spots are also evident, which is called the Auspitz’s sign [30]. The clinical manifestations of psoriasis vary considerably from patient to patient (Figure 1), from individual eruptions to generalized lesions, and the lack of a biological marker of disease makes it difficult to diagnose [25,26,27,30,31]. For this reason, different morphological forms of psoriasis are distinguished. These are, among others: plaque psoriasis (the common form of the disease; makes up about 90% of cases), guttate psoriasis (particularly common in children after strep throat infections), and pustular psoriasis (one of the most severe varieties of psoriasis, in which the spreading of pustules is generalized, with epidermal fulfillment and a severe general condition). Psoriatic arthritis also belongs to severe psoriasis types. In this case, the disease can cause arthritis, consequently leading to their deformation, impaired mobility and even disability. A form of psoriasis that can even lead to death is erythrodermic psoriasis, which is characterized by generalized inflammation of the skin of the whole body. Extreme inflammation and very severe peeling of the epidermis cause a disorder in body temperature regulation and the skin’s protective barrier. In addition to visible physical damage, psoriasis has a very negative effect on the mental state and quality of life of patients, and disrupts the daily functioning and interpersonal relations of patients [25]. Patients experience strong negative emotions associated with the appearance of the affected skin. These are, first and foremost, a feeling of blemish, expectation of rejection, guilt and shame, and sensitivity to the opinions of others. Disorders resulting from a lack of acceptance of the disease are depression and anxiety.

The cause of psoriasis remains unknown, but several theories exist. The genetic, immunological and numerous environmental factors influence the disclosure of the disease. Exogenous factors that trigger and exacerbate psoriasis symptoms include stress, excessive alcohol consumption, smoking, chronic bacterial and fungal infections, certain medications (e.g., corticosteroids, lithium salts, antimalarial drugs, beta-blockers, nonsteroidal anti-inflammatory drugs), hormonal changes during puberty or menopause, metabolic disorders, and physical factors such as injections, burns, insect stings, surgical wounds, cuts, scratches, and sunburn. The formation of psoriatic lesions at the site of skin injury is called the Koebner phenomenon. Genetic predisposition also has an important role in the development of the disease. It is estimated that approximately one-third of people with psoriasis have a positive family history associated with the disease. In addition, researchers have identified genetic loci associated with psoriasis. Confirmation of the participation of the genetic factor is the coexistence of psoriasis in twin brothers in 15–20% of cases and co-occurrence of psoriasis in monozygotic twins in 65–70% of cases [30]. Moreover, when one of the parents has psoriasis, the risk of a child getting psoriasis is around 20%, and when both parents have the disorder, the risk increases by up to 50–70%. These data suggest that both genetic and environmental factors influence disease development.

It is believed that psoriasis is caused by faulty signals in the body’s immune system that contribute to excessive proliferation of epidermal cells, abnormal keratinocyte differentiation, and inflammation [25,26]. A characteristic feature of psoriasis is the 8-fold shortening of the cell cycle and the survival of cell nuclei. In a healthy epidermis, cell migration from the basal to stratum corneum takes 28 days, while in psoriasis, 3–4 days [25]. Therefore, instead of being shed, the cells accumulate, causing visible psoriatic lesions [26]. The epidermis has a volume four to six times larger compared to healthy skin [25]. The number of keratinocytes is increased, and the individual cells are larger. Both DNA synthesis and mitotic activity are increased. It is thought that these changes stem from the premature maturation of keratinocytes induced by the activation of immune cells and the beginning inflammatory cascade in the dermis [27,32]. During psoriasis, abnormal infiltrating of T-cells, dendritic cells (DCs), macrophages, and neutrophils into the skin are observed [31]. It is well known that DCs present in the skin, such as Langerhans cells (LCs), play a major role in the early stages of psoriasis [33,34,35]. LCs can be activated by exo- or endogenous stimuli, such as trauma in the case of psoriasis. After activation, these professional antigen-presenting cells migrate from the skin into the draining lymph nodes, where they present antigens on their surface to T-cells, leading to activating them and the release of cytokines after binding [26]. T-cells are the major protagonists in the development of psoriasis [26,32]. These cells can be divided into two classes: CD4+ T-helper (Th) cells and CD8+ cytotoxic T-cells (Tc) [33]. CD4+ T-helper cells may be polarized to different fates depending on the needs of the immune response. DCs also can polarize T-cells to Th1 and Th17 fates. During inflammation, DCs produce a high number of cytokines such as interleukin (IL)-23, IL-12, and tumor necrosis factor (TNF)-α. Furthermore, increased levels of macrophages that also secrete IL-6, IL-12, and IL-23 and induce TNF-α can be observed in psoriatic lesions, which additionally amplify further reactions. IL-23 and IL-12 belong to the same family of cytokines, but they have different functions. While IL-12 is responsible for inducing interferon (IFN)-γ producing Th1 cells, IL-23 is responsible for inducing IL-17- and IL-22-producing Th17 cells. However, the activation of DCs in psoriasis is not entirely clear [34]. One hypothesis assumes the activation of DCs through recognition of AMPs. Dying cells release DNA, which, together with antimicrobial proteins produced by keratinocytes and infiltrating immune cells, form complexes that may act as an inflammatory stimulus by stimulating receptors on plasmacytoid DCs that in turn produce the cytokine IFN-α [27]. IFN-α then triggers activation, maturation, and differentiation of conventional DCs, which may stimulate T-cells through IL-23. These chemical messages trigger the secretion of IL-1, IL-6, and TNF-α by keratinocytes, which signal downstream immune cells to arrive. These immune cells move from the dermis to the epidermis and secrete proinflammatory cytokines, which stimulate keratinocytes to proliferate. Psoriatic autoimmune T-cells are biased to produce Th17-derived cytokines IL-17 and IL-22 [35]. In turn, these cytokines induce overexpression of AMPs in epidermal cells, indicating a positive feedback loop that further sustains innate activation of plasmacytoid DCs and leads to type I IFN-driven autoimmunity. In addition, type I IFNs directly upregulate the IL-22 receptor on keratinocytes, increasing their responsiveness to IL-22 cytokine, which inhibits terminal differentiation of the epidermal cells and induces hyperproliferation of keratinocytes leading to epidermal hyperplasia, a hallmark of psoriatic skin (Figure 2).

Nevertheless, Th17 cells play a main role in the pathogenesis of psoriasis [31]. Their pro-inflammatory effect and involvement in the integrated inflammatory loop with DCs and keratinocytes lead to increased production of AMPs, chemokines, and inflammatory cytokines enhancing the body’s immune response. IL-17A is primarily produced by Th17, but also by innate immune cells such as neutrophils, mast cells, and γδT cells. This cytokine is a direct potentiator in keratinocytes’ hyperproliferation and inhibits their differentiation. The data show that IL-17-stimulated keratinocytes produce chemokines and AMPs, which in turn promote the recruitment of Th17 cells and the production of more IL-17 by epidermal cells. Such a sequence of effects causes the formation of a positive pro-inflammatory loop, which intensifies inflammation in psoriasis. The increase in IL-17 level is associated with an impaired balance between Th17 cells and IL-23 produced by DCs and macrophages. IL-23 contributes to the enhanced proliferation of keratinocytes and is necessary for the survival and proliferation of Th17 cells, and therefore has a significant impact on the development of psoriasis. Currently, the IL-23/Th17/IL-17 axis appears to have a key role in initiating inflammation in this auto-immune-mediated disease [32].

## 3. Antimicrobial Peptides and Proteins

AMPs are a diverse, heterogenic group of small molecules (10–150 amino acids), which are subdivided by their chemical and biological characteristics [36]. They may have a net charge of −3 to +20, but most of them are positively charged, amphiphilic, hydrophobic (hydrophobic content less than 60%) molecules, present in all living organisms [2,23,36]. AMPs’ structures can be divided into several classes: α-helix, β-sheet, extended, or loop structure [2]. In the 1990s, AMPs were discovered to be abundantly expressed in human skin with broad antibiotic-like activity against pathogenic microorganisms such as Gram-positive and Gram-negative bacteria, yeast, fungi, protozoa, as well as viruses [2,23,37,38]. While AMPs are generally synthesized in the basal layer and the suprabasal layers (spinous and granulous layers), the active site for them is the cornified layer, where they are transported by lamellar bodies and play a vital role in the skin barrier [37,38]. AMPs may be produced both by cell types permanently residing in the skin and infiltrating immune cells, such as keratinocytes, eccrine sweat glands, sebocytes, mast cells, sweat ductal epithelial cells, neutrophils, natural killer cells, and phagocytes [23,38,39,40]. They may be synthesized by commensal skin microbiota too [39].

In healthy skin, some AMPs are constitutively secreted from resident cells onto the epidermal surface, where they serve as the first line of the host’s defense against pathogens [38,41]. In addition, some data suggest that the expression of AMPs in keratinocytes coincides with the expression of epidermal components among others: involucrin, loricrin, and transglutaminase I, which consist of the skin permeability barrier [39]. Upon injury, infection, chronic inflammation, or disruption of the skin barrier, the expression of AMPs increases to create a much stronger antibacterial shield [38]. Two mechanisms of AMPs’ antibacterial activity are proposed [39,41]. One of them assumes that skin commensal microbes produce antimicrobial proteins that have direct antibacterial activity by binding to negatively charged phospholipid groups and hydrophobic fatty acid chains of the microbial membrane, forming pores into the cell of the microbial membrane, and eventually leading to cell death. Another mechanism involves the activation of the cutaneous pathogen recognition system (e.g., Toll-like receptors or NOD-like receptors) in the presence of pathogenic microorganisms, which in turn triggers the release of AMPs from keratinocytes. Moreover, besides the ability to prevent pathogens’ adaptation, AMPs are able to neutralize endo- and exotoxins produced by microbes. In vitro studies [38] show that factors such as pH, oxidized or reduced forms of AMP, and recombinant or natural AMP may influence antimicrobial activity, so it is difficult to predict and interpret in vivo effects of AMPs.

In recent years, the novel functions of AMPs, far beyond their antimicrobial activity, were discovered [2,23,38,39]. It is shown that they can modulate host inflammatory responses through a variety of mechanisms including cytokine and chemokine production, acting as chemotactic agents, regulators of cell proliferation and differentiation, angiogenic factors, and proteinase inhibitors. Furthermore, AMPs are able to serve as interplayers between innate and adaptive immune systems by triggering an antigen-driven immune response, attracting immune cells to the site of injury, infection, or inflammation, and modulating Toll-like receptor responses. AMPs may activate mammalian cells through several mechanisms including direct binding to specific receptors, which leads to the initiation of receptor signaling, altering the receptors’ membrane domains without serving as a ligand, and stimulation of membrane-bound growth factors [2,23]. The wide spectrum of functions described by AMPs underlines the importance of their correct regulation for maintaining an optimal skin barrier [38].

Abnormal expression of AMPs is observed in several inflammatory skin diseases, particularly psoriasis [2,23,37,38,39]. Increased levels of certain AMPs during disease might contribute to inflammation potentiating, and as a result, contribute to psoriasis itself [38]. In research on the pathogenesis of psoriasis, three subclasses of AMPs have gained the greatest interest as key players in the development of the disease: defensins, cathelicidin, and S100 proteins [2,23,34]. The understanding of the novel functions of AMPs and their role in the pathophysiology of inflammatory skin diseases may contribute to the extension of conventional treatments and the use of AMPs for therapeutic purposes [39].

## 4. Therapeutic Potential of AMPs

Humans have the capacity to produce a wide variety of AMPs that form the host’s first line of defense against infection and generally play an important role as effector factors of the innate immune system [5,6,7,8,42,43]. AMPs show strong activity (even in low concentrations in vitro) against many pathogenic microorganisms and have anti-inflammatory properties, making them important compounds (e.g., Granulysin-derived peptides, CEN1HC-Br, LL-37, MX-594AN) for the development of new therapeutic agents in infectious and inflammatory diseases [44,45,46,47,48,49,50,51,52,53]. However, it should be mentioned that the effects of AMPs on the immune system are extensive and complex, and in addition to their anti-inflammatory properties, many AMPs exhibit strong pro-inflammatory properties, reflecting the need to tightly control any immune response [54,55]. Due to its dual role in autoimmunity, the use of AMP as a therapeutic target or as a means of preventing and treating autoimmune diseases should be considered with caution. It is known that abnormal production of AMPs produced by neutrophils or epithelial cells promotes inflammation by favoring an autoimmune response. Activated neutrophils in tissue produce extracellular neutrophil traps (NETs), which are composed of their own nucleic acids from the nucleus associated with granular cytoplasmic proteins rich in AMPs [56,57]. Abnormal NET production and their impaired clearance would stimulate pDCs through TLR7 and TLR9 to produce type I IFNs, which are important contributors to autoimmune diseases [58,59,60,61]. In contrast, recent studies have shown that AMPs produced by specific non-immune cells have immunoregulatory properties against various types of innate and adaptive immunity, leading to the induction of Treg cells and thus preventing the development of autoimmune disease [62]. An interesting example of AMPs showing both anti- and pro-inflammatory properties in autoimmune disease is the effect of cathelicidin in type I diabetes (T1D). The role of this protein in the development of T1D has been well-demonstrated in a mouse model by Diana et.al [63]. The first study showed that cathelicidin is involved in disease initiation in young mice with diabetes without obesity [63]. Neutrophils transiently infiltrated the pancreas and produced cathelicidin in complex with their own DNA and anti-DNA immunoglobulin G. These complexes successively activated pDCs via TLR9, inducing the production of type I IFNs, which promoted the development of T1D [64]. On the other hand, a follow-up study of the same group showed a protective role of cathelicidin against T1D. The results confirmed that cathelicidin is normally produced by pancreatic β cells in adult non-autoimmune mice, but not in non-obese diabetic mice. Conversely, treatment of adult mice with diabetes without obesity in a pre-diabetic state with recombinant cathelicidin induced regulatory macrophages and T cells in the pancreas, preventing disease progression. In psoriasis, LL-37 controls inflammation by acting as both a pro- and anti-inflammatory agent. LL-37’s pro-inflammatory activities include down-regulation of IL-10, up-regulation of IL-1β, IL-12p40, and IL-18, induction of type I IFN in pDCs and keratinocytes, and mast cell degranulation and release of inflammatory mediators [60,65,66,67,68]. The anti-inflammatory properties of LL-37 involve inhibition of AIM2 inflammasome formation and suppression of IFN-γ, TNF-α, IL-4, and IL-12 [46,60]. Furthermore, the presence of AMP (e.g., LL-37) during the differentiation of macrophages and DCs may influence their polarization toward a pro-inflammatory M1 phenotype [69], whereas LL-37 alters the differentiation of DCs such that they promote Th1 cell responses and thereby promote enhanced adaptive immunity [70]. LL-37 can neutralize the pro-inflammatory activity of selected TLR-ligands such as lipopolysaccharide [71] and, at high concentration, can contribute to the local regulation of inflammation by inducing the expression of the anti-inflammatory cytokine IL-10 in monocytes/macrophages, DCs, and B and T cells [72]. AMPs such as hBD-2, LL-37, S100A7, and HNP induced in acne lesions can increase inflammation by recruiting and activating immune cells and releasing pro-inflammatory mediators [73,74,75]. Given that *P. acnes* causes inflammation in acne and that AMPs such as hBD-2, LL-37, S100A7, and RNase 7 show killing activity against *P. acnes* and inhibit inflammatory reactions mediated by bacterial products, these molecules are potential candidates for the prevention and treatment of acne vulgaris. One hypothesis explaining the dual role of AMPs in autoimmune diseases is that the immune function of AMPs is related to post-translational modifications of peptides, such as carbamylation or citrullination. Indeed, such modifications of cathelicidin reduce its positive charge, increase its chemotactic activity, and alter its ability to bind nucleic acids, thus reducing its pro-inflammatory potential. In addition, modified AMPs can be a source of autoantigens, but not their native forms. Importantly, modifications of susceptible proteins that occur in inflammatory conditions, such as activated neutrophils, may represent a general mechanism for controlling the inflammatory response. Undoubtedly, finding the balance between anti-inflammatory and pro-inflammatory responses mediated by AMPs will be the focus of future drug discovery work using them. In fact, AMPs are being considered as potential therapeutics. This is very important, especially because of the rapid increase in multidrug resistance of microorganisms, which is prompting the development of improved therapies based on “natural antibiotics”. Currently, many AMPs are being evaluated in late-stage clinical trials, not only as novel anti-infective therapeutic agents, but also as innovative product candidates for immunomodulation, wound-healing support, and prevention of surgical scarring [5]. Still, despite the myriad of AMP-related patents filed, only a very narrow proportion of the AMPs have already been approved, either by the US Food and Drug Administration (FDA) or EU European Medicines Agency (EMA) [6]. Some of the AMPs which have already received the certification or are in clinical trials are reported here in Table 1.

There are many advantages to using AMPs as therapeutic agents. First, AMPs are endogenous proteins, which eliminates the risk of potential allergic reactions [8]. AMPs have existed for millions of years as an important component of host defense systems, yet still show high activity against many bacteria, suggesting that acquiring complete resistance is unlikely [7,8,42,43]. The faster mutation rate in bacteria compared to the rate of adaptive mutation in mammals suggests that mutations that may cause resistance have already occurred [8]. Nevertheless, it has been shown that some bacteria, such as *Staphylococcus aureus*, have developed mechanisms leading to reduced sensitivity to AMPs [76]. In addition, some resistant bacteria may interfere with the direct antimicrobial activity but not the inflammatory properties of AMPs [7]. A recent systematic study [77] of the evolution of *Escherichia coli* resistance to 14 chemically differentiated AMPs and 12 antibiotics suggested a very low level of resistance to AMPs through point mutations and gene amplification, while antibiotic-resistant bacteria showed no cross-resistance to AMPs. Therefore, the possibility of developing resistance to AMPs in antimicrobial species is severely limited. In addition, some AMPs act as anti-biofilm agents against many species of bacteria [7,42,78]. An example is IDR-1018, which showed a broad-spectrum anti-biofilm activity, driven by an important signal in the development of biofilm, which is bacterial guanosine pentaphosphate [79]. 

Another human antimicrobial peptide, LL-37, prevents biofilm by down-regulating key biofilm genes, inhibiting the initial attachment of bacteria, and stimulating twitch motility [80]. Although AMPs and antibiotics have shown comparable anti-biofilm activity against methicillin-resistant *Staphylococcus aureus* (MRSA), *Staphylococcus epidermidis*, *Pseudomonas aeruginosa*, and *Escherichia coli* [81], they have been shown to act synergistically with various antibiotics and are effective in animal models of biofilm infection, suggesting a promising role for them in preventing colonization and developing treatments for biofilm-associated infections [7,42]. The broad-spectrum activity of AMPs, high selectivity, and relatively low likelihood of resistance emergence are advantages of AMPs as drugs, resulting in peptides showing higher success rates in all stages of clinical trials compared with small-molecule drugs [5,42]. This likely reflects fewer safety concerns about peptides since AMPs provide the opportunity for therapeutic interventions that closely mimic natural pathways [5]. Many reports have suggested that AMPs have a low propensity to cause toxicity when administered both topically and parenterally [7]. Furthermore, no serious adverse effects have been reported to date in animals treated with AMPs. The current use of several AMPs-based drugs in humans is to supplement the peptide at sites in the body where endogenous levels are insufficient, and therefore there is very little likelihood of adverse side reactions [5]. The excellent safety profiles of AMPs can also be explained by the degradation products of peptides and antimicrobial proteins that are natural amino acids [82]. One of the biggest problems with conventional antibiotics is the potential for septic shock, as has been reported with ciprofloxacin, for example [83]. However, it has been reported that some AMPs prevent endotoxin-induced sepsis both in vitro and in a mouse model [84]. Topical treatment, which is the most common route of administration for AMPs, further reduces the risk of systemic adverse events because only a small portion of the peptide reaches the bloodstream [85]. It is also worth noting that therapeutic peptides are considered less immunogenic than recombinant proteins and antibodies [86]. Although many AMPs are less toxic to eukaryotes, the systematic toxicity of their use has not been evaluated, so applications of AMPs in clinical trials are mainly limited to topical applications [42] with some exceptions, where they have been administered intravenously. Latter AMPs include, for example, p2TA (AB103) screened for treating patients with necrotic soft tissue infections receiving standard therapy [87], as well as Neuprex(rBPI21) evaluated for medical applications in meningococcemia and prevention of infectious complications after post-traumatic hemorrhage [88,89]. Another compound administered intravenously in clinical trials is Ghrelin, which is the only known circulating peptide in the stomach that stimulates appetite. Studies with intravenous Ghrelin include testing its use to control alcohol craving and use [90], evaluating the effects of Ghrelin administration on dopamine signaling in humans [91], examining the effects of short-term Ghrelin administration to humans on metabolism [92], including assessing the effects of acute Ghrelin administration on insulin resistance as measured by the hyperinsulinemic euglycemic clamp, and evaluating the effects of Ghrelin on local and systemic lipolysis. A peptide containing the first eleven residues of human lactoferrin (hLF1-11), administered intravenously at a dose of 5 mg, also underwent clinical trials [93]. The goal of the study was to develop hLF1-11 as an effective and safe antibacterial and antifungal agent for the treatment of fungal and bacterial infections that develop during neutropenia resulting from myeloablative therapy to prepare recipients for hematopoietic stem cell transplantation.

Although AMPs show high therapeutic potential, they still have several undesirable properties for clinical applications [5,6,7,42,43]. Natural peptides are generally unstable in the gastrointestinal tract and other body fluids, have poor absorption and distribution, and rapid metabolic degradation and excretion result in low bioavailability, which is one of the greatest challenges in successful clinical applications of AMPs after oral administration [5,6,42]. Peptides are unstable in the gastrointestinal tract due to both acute gastric pH and the presence of proteolytic enzymes such as trypsin and pepsin [5,42]. Furthermore, the intestinal permeability of AMPs is limited by their high polarity and molecular weight [5]. Consequently, although there are examples of AMPs that are intended for oral or intravenous administration, it is recognized that topical application remains the most common route of AMP delivery. It is worth noting that even after topical treatment, AMPs are susceptible to degradation by tissue proteolytic enzymes. Systemic administration of AMPs is also limited due to rapid proteolysis in the bloodstream, coupled with rapid withdrawal from the circulation by the kidney and liver (renal and hepatic clearance, respectively), leading to short in vivo half-lives, cytotoxic profiles in the blood, and protease degradation [5,7,42]. This labile nature allows the body to rapidly modulate peptide hormone levels to maintain homeostasis [5] but poses challenges to therapeutic development, although it appears that stability issues can largely be overcome by the addition of protease inhibitors [94]. Perhaps the most important reason why so few AMPs have been clinically tested to date appears to be their possible cytotoxicity [43]. AMPs can, in parallel with their affinity for the microbial membrane, interact with and degrade the membrane of eukaryotic cells, especially erythrocytes, leading to hemolysis [5]. For example, mammalian protegerin-1 AMP reduced bacterial numbers in a mouse model of sepsis but also reduced survival [95]. This was likely due to the induction of endotoxin release and consequent modification of the host’s innate immune response. Despite the problems with the cytotoxicity of AMPs, they are certainly less important in the topical treatment of skin diseases than in the systemic treatment of infectious or inflammatory diseases [43]. In this context, however, it should be mentioned that although tyrothricin can only be administered topically because of its systemic toxicity, it is still effective and well-tolerated in the treatment of skin diseases. Recently, strategies to reduce the cytotoxicity of AMPs have been increasingly developed [5,6,7,42,43]. To this end, many attempts have been made to decipher the structural features and physicochemical properties of AMPs. It is known that increased hydrophobicity and/or amphipathy of AMPs can lead to impaired hemolysis [96,97]. Nevertheless, the relationship between hydrophobicity/amphipathy and risk of cytotoxicity appears to be complex, and the outcome does not depend on only one factor, but rather requires a balance between charge, hydrophobicity, and amphipathy. Moreover, the flexible structures of AMPs may also induce interactions with unintended components, which may result in undesirable side effects [7,42]. Many methods have been proposed to improve the stability, safety, and efficacy of AMPs [5,6,7,8,42,43]. The use of delivery vehicles or nanocarriers to adsorb or encapsulate AMPs can specifically target delivery to a specific site with the controlled release over time. In addition, AMPs are also chemically modified to increase their durability and efficiency.

The cationic and amphiphilic properties of AMPs are related to their biological activities, which in turn are closely related to their chemical structures [98]. In general, only a few key amino acids in AMPs are critical for antimicrobial activity, while many other residues can be replaced without affecting their function [99]. Several amino acids have been extensively studied for their significant effects on the structure or activity of AMPs. Proline (Pro) and glycine (Gly) provide greater flexibility in peptide structure, which may help induce hemolytic activity. Tryptophan (Trp) has been shown to facilitate interactions with bacterial membranes when placed at the interface between the hydrophobic and hydrophilic sides of an amphiphilic helix. Cysteine (Cys) is an amino acid with sulfur in the side chain, which makes it highly reactive. It is easily oxidized to form a dimer, forming a disulfide bond between two Cys residues. These disulfide bonds are highly hydrophobic and are important in the structure of many natural AMPs. It also increases resistance to proteolytic degradation. Therefore, amino acid substitutions are widely used in the design and optimization of AMPs as a specific class of therapeutic agents. Before modifying a lead peptide drug candidate, it is necessary to determine the minimally active sequence with the desired biological properties, and then make further modifications to the exchangeable residues and C- and N-terminus of the lead peptide to generate the final peptide drug [100]. However, the main challenges in developing AMPs as systemic drugs are their pharmacodynamics and in vivo instability. To increase their stability and effectiveness and reduce their cytotoxicity and non-targeted side effects, some key strategies [96,99,100,101] are being employed, which include cyclization, incorporation of atypical amino acids into the AMP sequence, side/end chain modifications, peptide conjugation, nanoparticle formulations, or peptidomimetics whose major components mimic the peptide structure. Cyclization is a common peptide modification technique used to achieve improved antimicrobial activity and stability and reduced hemolytic activity [101]. Cyclization can involve various strategies, such as head-to-tail cyclization, backbone-to-side chain cyclization and side-chain-to-side-chain cyclization [102,103,104]. Peptide cyclization can enhance proteolytic stability and cell permeability and allows the secondary structure of the peptide to be mimicked and stabilized [105,106]. Without combining with other peptides, a single peptide sequence cannot form loop or twist structures, but cyclization facilitates the formation of these secondary structures by pre-organizing intramolecular interactions [100,107]. Many studies have shown that cyclic peptides generally exhibit beneficial properties as antimicrobial agents, which is attributed to increased proteolytic stability and conformational rigidity [108]. In addition, cyclization also increases cell selectivity, leading to reduced host cytotoxicity [109,110]. Macrocyclic peptides are common among naturally occurring AMPs, some of which have been approved by the FDA and used in clinical practice [108]. In addition, peptide macrocycles have attracted considerable attention as potential AMPs due to their synthetic availability. Both the peptide sequence and the nature of cyclization are important parameters for optimizing antimicrobial activity and human toxicity. The potential benefits of macrocyclic peptides are driving recent research to improve oral bioavailability, which is typically poor for peptide drugs. In general, cyclic peptides can be formed by disulfide bonds (heterodisulfide cyclization by a non-amide bond) and/or amide bonds (homodisulfide cyclization by an amide) [102,111,112,113,114]. Heterodisulfide cyclization occurs by oxidation of the Cys side chain thiols, which forms single or multiple disulfide bridges, and is the most commonly used method in AMP design and optimization [115,116]. Cyclized peptides with single or multiple disulfide bridges are likely to form a secondary or higher structure than their linear form, thus reducing their susceptibility to proteolytic degradation and increasing their stability. Cyclization through disulfide bonds occurs naturally in AMPs such as the human β-defensin family (hBDs) [117]. Several systemically active AMPs, such as bacitracin, polymyxin B and E, tyrotricin, gramicidin S, and daptomycin, are also cyclic peptides [116,118,119]. Inspired by the intramolecular disulfide bonds between cysteine residues in natural cyclic hBDs, Mwangi and colleagues [120] successfully engineered an AMP that was cyclized by forming disulfide bonds. Two cysteine residues were introduced into cathelicidin-BF15-a3, a peptide derived from the venom of the snake *Bungarus fasciatus*. The cyclic peptide showed excellent activity against *P. aeruginosa* and *A. baumannii*, low hemolysis, high in vivo stability, and low propensity to induce resistance. The second strategy for the stable formation of cyclic AMP is lactamization, which is the formation of intramolecular peptide bonds to produce homodietic cyclic peptides [121]. Some bacteria can produce cyclic peptides containing lanthionine and/or methyllanthionine amino acids with thioether bridges [122,123]. Peptides with intramolecular thioether bridges are more stable than peptides with disulfide bridges. Newer approaches to stable cyclic peptides utilize a wide range of commercially available, properly functionalized non-naturally occurring amino acids to enable intramolecular click reactions [102,111,112,113,114,124]. These chemoselective cyclizations include, but are not limited to, Michael-type (thiolene) additions, 1,3-dipolar (azide-alkyne) Huisgen cycloadditions, and ring-closing olefin metathesis, also known as the Grubbs reaction. The cyclization of peptides by ring-closing metathesis, leading to so-called stapled peptides, is gaining traction toward the development of synthetic AMPs and structurally related peptides, such as cell-penetrating peptides (CPPs), by presenting a restricted and metabolically stable helical conformation, commonly known to be essential for the membrane interaction ability of most cationic amphipathic AMPs and CPPs [124,125,126]. Another possible chemical modification that can enhance the proteolytic stability and antimicrobial efficacy of AMPs is the inclusion or replacement of natural amino acids with amino acid analogs or unnatural amino acids [99]. Unusual or unnatural amino acids refer to those amino acids that have been chemically modified or are found in living organisms but are not found in proteins. A popular approach to improving the bioavailability of peptides has been to replace proteinogenic amino acids in the native sequence with non-canonical amino acids, such as D-amino acids, N-methylated amino acids, and other non-canonical residues [99,127]. Optical activity is one of the most important biochemical characteristics of amino acids. Most amino acids, apart from glycine, have one or two asymmetric atoms, allowing them to form two enantiomeric L- and D- forms, which exhibit different biological activities [128]. L-amino acid peptides are susceptible to rapid proteolysis; however, D- or other atypical chemically modified amino acids can reduce the peptides’ susceptibility to proteolysis, since altering the stereochemistry of the AMP can render it unrecognizable by proteolytic enzymes [129,130]. For example, the D-amino acid variant of pleurocidin is resistant to degradation by trypsin, plasmin, and carboxypeptidase [131]. Therefore, the inclusion of D- or atypical amino acids or the replacement of proteinogenic amino acids with non-proteogenic residues offers the hope of improving the stability and/or bioavailability of peptides. Merrifield and colleagues [132] pioneered the use of D-amino acids to improve the bioavailability of AMPs by producing “retro” (inverted sequence, all L-amino acids), “enantio” (normal sequence, all D-amino acids), and “retro-inverso” (or “retro-enantio”) hybrid analogs of AMPs, the latter being equivalent to their original membrane-active AMPs matrices while exhibiting much higher proteolytic stability. The use of retro-inverso analogs of bioactive peptides has persisted as a popular approach to enhance AMP stability against proteolysis since the main target of amphipathic α-helical AMPs is the bacterial membrane, which does not involve stereospecific interactions [133]. Hence, resorting to retro-inverso analogs, or simply discrete L- amino acid D substitutions, remains an interesting option for improving AMP performance [134]. In general, natural AMPs and their enantiomers composed of D-amino acids exhibit comparable antimicrobial activity [131,132]. Nevertheless, if D-amino acids are incorporated into an AMP agent, the helical structure can be disrupted, resulting in a complete loss of activity [135]. In addition, partial substitution of D-amino acids can lead to poor side chain packing, resulting in reduced hydrophobicity and further reduced activity [136]. Introducing unnatural amino acids into the peptide sequence is an effective strategy for extending the half-life of peptide drugs. An example is selepressin, which is derived from vasopressin and has similar target selectivity but a longer plasma half-life [137,138]. In addition, Arias and co-workers showed that lysine substitution for 2,3-diaminopropionic acid (four to one methylene unit) in Trp-rich peptides increased their antimicrobial efficacy 4-fold against E. coli, probably due to increased membrane permeabilization [139]. Oliva and colleagues designed cationic synthetic peptides containing unnatural amino acids such as 2-naphthyl-L-alanine and S-tert-butylthio-L-cysteine, which exhibit high activity against a broad spectrum of pathogens and enhanced proteolytic stability [140]. LTX-109, a nasal decolonization agent for methicillin-resistant and susceptible *Staphylococcus aureus*, is another example of a tripeptide consisting of a lipophilic unnatural tryptophan residue flanked by two arginine residues, which highlighted how effective the use of unnatural amino acids can be for developing therapeutic AMPs [141,142]. Post-translational modifications play an important role in the function of AMPs and are most commonly used in peptide design [101]. N-/C-terminal modifications (terminal capping) provide simple but useful approaches to increasing peptide stability and efficacy in vivo [99]. Typical modifications include amidation (C-terminal), acetylation (N-terminal), and methylation (N-terminal), which are often used to increase resistance to peptidases and proteases. For example, a cytotoxicity assay showed that N-terminal acetylation and C-terminal amidation of β-hairpin AMP tachyplesin I exhibited higher toxicity against both cancer cells and normal human cells with increased hemolytic acidity [143]. However, the modified tachyplesin I was resistant to proteolytic degradation in human serum compared to the original molecule. In addition, AMP derived from human apolipoprotein B, which underwent simultaneous N-acetylation and C-amidation, showed more than 4-fold increased proteolytic stability compared to unmodified AMP after incubation with 10% fetal bovine serum for 1 h [140]. It is worth noting that AMPs with C-terminal amidation are common in nature, and N-terminal acetylation is a protein modification frequently observed in eukaryotic and prokaryotic cells. Although N-terminal acetylation blocks the activity of aminopeptidases, thereby improving the proteolytic stability of the peptide, it reduces the net positive charge by one, which can reduce antimicrobial activity [108]. In contrast to N-terminal acetylation, C-terminal amidation has been shown to improve the antimicrobial efficacy of many membrane-disrupting AMPs, possibly due to increased α-helix stability at the peptide–membrane interface, allowing greater membrane disruption and pore formation [110,144,145,146,147]. An example of natural AMPs in which the modifications in question occur is the temporin family, in which most peptides are amidated at their C-terminus [148,149]. It has been shown that the C-terminal amide group of maximin H5 can increase antimicrobial efficacy without increasing lytic capacity, while the N-terminal lipidated C 4 VG analog of 16KRKP shows enhanced antimicrobial activity against various Gram-negative bacteria [150]. The goals of N-terminal lipidation are to increase the neutralization of LPS, increase the stability of proteases and peptidases, and reduce cytotoxicity [151]. In addition, labeling of hydrophobic ends is a common method for enhancing the activity of antimicrobial peptides [101]. In addition to end modifications, side chain modifications also make peptides resistant to proteases. For example, alkylation of the amino group of the Lys side chain is a useful strategy for preserving and enhancing antimicrobial activity and reducing susceptibility to enzymatic degradation [139,152]. Halogenation, on the other hand, is strongly associated with AMP activity, specificity, and stability. An example is the AMP jelleny-I, in which halogenation was introduced by replacing phenylalanine with a halogenated phenylalanine analog, enhancing in vitro antimicrobial activity and activity against biofilm [153]. With halogenation, the proteolytic stability of jelleny-I is increased 10–100 times. Halogenation is also associated with AMP specificity. O-fluorine substitution at phenylalanine residues maintains the activity of Temporin L on *E. coli* but leads to loss of activity on *S. aureus* and *P. aeruginosa* [154]. In addition, PEGylation and glycosylation of amino acid side chain groups are used to improve the bioavailability of peptides [124,155,156,157,158]. PEGylation can also protect peptides from proteolytic degradation, reduce immunogenicity by limiting their uptake by dendritic cells, and reduce renal clearance, thereby improving the half-life of peptides in the circulation for intravenous administration by up to more than 2 orders of magnitude [124,155,156]. However, PEGylation of AMP can also reduce the binding of peptides to the bacterial membrane, thereby reducing their antimicrobial efficacy, despite the beneficial reduction in cytotoxicity and hemolysis [108]. For example, PEGylation of magainin significantly reduced the peptides’ cytotoxicity, but also resulted in reduced activity against *E. coli* and *S. epidermidis* compared to native peptides [159,160]. Another approach that was used was chemo-reversible PEGylation of arginine side chains via cleavable phenylglyoxal linkers, which produced arginine-rich AMPs whose resistance to serum proteases was dramatically increased, while allowing slow release of the parent bioactive peptide over several hours or days [161]. Chemo-reversible PEGylation of arginine side chains via cleavable phenylglyoxal linkers was also used to produce arginine-rich AMPs whose resistance to serum proteases was dramatically enhanced, while allowing slow release of the parent bioactive peptide over several hours or days [162]. Glycosylation is a widely used approach in post-translational protein modifications [157,163]. Nowadays, technology makes it possible to tailor glycan modifications to improve stability and antimicrobial activity and specificity. Naturally glycosylated AMPs derived from insects and rich in proline have inspired chemists and bioengineers to develop artificial site-specific glycosylation methods to deliver glycopeptides with improved stability, bioactivity and specificity. In addition, increased bioavailability of peptides can be achieved by coupling to large and relatively stable proteins. For example, when conjugated to human albumin, the peptide infestin-4 shows a significant increase in circulation time after intravenous injection due to decreased renal clearance [164]. Alkylation of the ε-amino group of Lys side chains is an equally simple approach that has been used in AMPs to preserve or even enhance their antimicrobial activity while possibly reducing their instability for enzymatic degradation [139,152]. Peptide conjugation has been the goal of most research in recent years to produce active and stable AMPs with high selectivity [101]. In addition to repeating the same amino acid motifs, different side chains or fragments of AMPs can be used. For example, coupling fatty acids of 8–12 carbon atoms to the 4 or 7 side chain of the D-amino acid Ano-D_4,7_ improves antimicrobial selectivity and anti-biofilm activity [165]. In addition, the new peptide shows high stability against trypsin, serum, salt, and different pH environments. Conjugation of different AMPs can also be carried out. For example, a hybrid peptide (PA2-GNU7) constructed by adding PA2 to GNU7 has high activity and specificity to *P. aeruginosa* [166]. On the other hand, the combination of LPS-binding peptide (LBP)14 with marine AMP-N6 showed enhanced killing activity against *E. coli* MDR and the ability to neutralize LPS both in vitro and in vivo [167]. Conjugation of R9 with magainin or M15 with three glycines increased antimicrobial activity 2 to 4-fold against Gram-positive bacteria such as *S. aureus* and *E. faecalis*, and 4 to 16-fold against Gram-negative bacteria such as *E. coli* and *P. Aeruginosa* [168]. Further strategies include the use of peptidomimetics, which are peptide-like polymers made from a backbone that is altered from the peptide [96]. The main concept behind the use of peptidomimetics is to maintain activity by preserving the 2D and 3D spatial arrangement of the side chains but modifying the backbone to prevent proteolysis. The backbones of peptidomimetics are not entirely based on α-amino acids, but also on β-amino acid oligomers, arylamide oligomers and phenylenetinylenes [169]. Peptidomimetics generally mimic the structure, activity and selectivity of antimicrobial peptides while improving plasma half-life [7]. Some examples of peptidomimetics include peptoids, ceragenins, oligoacylins and β-peptides [170,171]. Peptoids are isomers of peptides in which the side chain is linked to the backbone nitrogen instead of the α-carbon, or poly-*N*-substituted glycine in which the side chain is linked to the amide nitrogen instead of the α-carbon in the main chain, making them resistant to protease degradation [172]. Examples of such AMPs include the cationic peptide SA4 and its poly-*N*-substituted glycine homolog SPO, which inhibit plankton and biofilm formation of strains of *A. baumannii*, which are susceptible to multidrug resistance [173], and SMAMP10, which is a potential drug for intravenous treatment, does not cause drug resistance and has potent inhibitory activity against MRSA and vancomycin-resistant *Enterococcus faecium* [174]. In addition, pexiganan-derived peptides have been shown to mimic not only the 1D structure, but also the 2D structure, function and mechanism of action of pexiganan [175]. Circular dichroism studies confirmed that peptoids adopt an α-helical structure in the presence of phospholipids, while X-ray reflectivity showed that peptoids bind to the membrane and are active in the membrane. Cyclization of peptoids also improves membrane permeation properties, leading to better antimicrobial agents [176]. Another promising approach in peptidomimetics is the combination of native peptide and pseudopeptide structures, resulting in hybrid structures [177,178]. Nanotechnology provides delivery strategies for AMPs, reducing their toxicity and increasing their stability and target selectivity, as nanoparticles are excellent target-specific drug delivery systems [179]. Nanoscale materials have a large specific surface area, flexible surface functionalization and unique physicochemical properties [180,181]. In addition, some nanoparticles, such as silver and titanium dioxide, exhibit antimicrobial activity on their own, so many nanomaterials have been developed as potential antimicrobial agents [99]. AMPs are sensitive to proteolytic enzymes, limiting their use, which can be enhanced with nanoparticles. After exposure to proteolytic enzymes, nanomolded PA-13, which was electrostatically encapsulated in nanoparticles, retained its killing activity against *P. aeruginosa* in both in vitro culture and an ex vivo skin model in the pig [182]. However, unencapsulated PA-13 lost its antimicrobial activity. In addition, nanoconstruction can be used to design non-toxic AMPs. There are many forms of nanoparticles used to deliver AMPs; however, this issue is not the focus of this review.

Excessive or decreased expression of AMPs is often observed in infected or inflamed skin, possibly due to local induction by microorganisms and pro-inflammatory cytokines [8,42,43]. For example, there is an association between reduced AMP levels in burns, atopic dermatitis, and chronic wounds compared to normal skin. Conversely, almost all known AMPs, such as HBD-2 and -3 β-defensin, LL-37 cathelicidin, psoriasin (S100A7), RNase 7, ALP/SLP1, Elafin/SKALP, and others, are highly upregulated in psoriatic skin, which explains why psoriasis patients suffer from fewer bacterial skin infections than might be expected. These findings indicate that AMP has potential for therapeutic use. The current state of research is to better understand the physiological role of AMP in human skin, which is the first step in today’s dermatology approach [43]. Based on these findings, targeted therapy strategies have been developed that focus on the mechanism of action. Ultimately, AMPs that demonstrate the ability to modulate specific disease-related processes will have to be tested in clinical trials for their effectiveness and tolerability. The results of these studies hold promise for effective treatments for human skin conditions such as psoriasis.

Efforts to translate AMPs-based research results into pharmaceutical candidates are expected to accelerate in the coming years due to technological advances in many areas, including a better understanding of AMPs’ mechanism of action, intelligent formulation strategies, and advanced chemical synthesis protocols [5,6,7,42,43]. Still, the relatively costly production of AMPs, along with regulatory hurdles for new antimicrobial products, remains a major barrier to the commercialization of AMP. Despite this, intensive work is being carried out around the world to introduce new AMPs or their derivatives into clinical applications in the treatment of infectious or inflammatory skin diseases.

## 5. S100 Proteins

Very well-known AMP-family members are S100 proteins that constitute the largest, multigenic, and calcium-binding protein family in vertebrates [11,183,184,185]. To date, over 20 types of these proteins have been identified, of which 13 are expressed in the normal or diseased human epidermis [2,186]. The name of the S100 proteins is due to their biochemical characteristics, namely, they are 100% soluble in saturated ammonium sulfate at neutral pH [184]. S100 proteins are small, acidic proteins with a molecular weight of 9–13 kDa [2,23,185]. They are produced as monomers, but exist in cells as anti-parallel homo- and heterodimers, in which monomers are held together by non-covalent bonds and are oriented by a two-fold axis of rotation [2,11,185]. Dimers can further associate to form higher-order multimers [187]. Each S100 monomer consists of two helix–loop–helix structural motifs that are Ca^2+^-binding domains termed EF-hands [11,184,185]. We distinguish the non-canonical N-terminal (S100-specific or pseudo-EF-hand) EF-hand motif composed of 14 amino acids and the canonical C-terminal (calmodulin-like) EF-hand motif composed of 12 amino acids [184,187]. The EF-hands structural motifs are linked with a flexible hinge region (Figure 3) of variable length (12–14 amino acids) that exhibits the most diverse amino acid sequence within S100 proteins and is critical to target protein binding [184]. Upon Ca^2+^-binding, almost all S100 proteins undergo a conformational rearrangement that reorients helix III to expose a previously covered hydrophobic cleft, which is the target protein recognition site [11,184,185]. Because S100 proteins exist in the form of dimers, they can bind two homologous or heterologous target proteins at opposite ends of the dimer forming a heterotetramer complex [185,188].

S100 proteins are expressed in a cell- and tissue-specific manner and are involved in multiple intracellular and extracellular functions [183,184]. To activate these proteins, calcium-binding and dimer formation is required. In some cases, also other metal ions, including zinc or copper, can be banded by S100 proteins and contribute to their activation. Mustafa et al. [189] demonstrated that melatonin also can increase the expression of S100 proteins in scrotal skin. Moreover, Hussein et al. [190] showed that melatonin increases the S100 proteins’ expression in the connective tissue capsule and the adrenal cortex of Soay rams. On the other hand, melatonin can inhibit expressions of S100β protein in the hippocampus of rats with senile dementia [191]. Vitamin D analogs are known as anti-psoriatic compounds that reduce psoriasin (S100A7) and koebnerisin (S100A15) expression in psoriatic skin [3,192]. In addition, S100A7 overexpression in the plaques of patients with psoriasis vulgaris with joint inflammation compared with psoriasis vulgaris was demonstrated [193]. Based on their roles in biological processes, S100 proteins may be subdivided into three main subgroups: proteins possessing only intracellular functions (e.g., S100A1), proteins having both intracellular and extracellular effects (e.g., S100B), and S100 proteins mainly exerting extracellular functions (e.g., S100A15). S100 proteins are involved in, among others: transducing changes in intracellular calcium concentration to biological responses acting as calcium sensors, regulating enzymes, modulating cell metabolism, interacting with intracellular receptors, transcriptional regulation, cytoskeletal membrane interaction, and regulating protein phosphorylation [2,11,19,23,183,184,187,188,194]. When secreted to the extracellular space, S100 proteins may regulate cell growth, proliferation, differentiation, and apoptosis. S100 proteins may also act as damage-associated molecular pattern (DAMP) molecules, which have a critical role in the regulation of immune homeostasis [183]. Damaged or stressed cells and activated phagocytes (neutrophils, macrophages) secrete DAMPs that act as danger signals inducing an inflammatory response. After binding to several cell surface receptors, including the receptor for advanced glycation end products (RAGE), Gi-protein-coupled receptor (GiPCR) as well as Toll-like receptor (TLR) 4, S100 proteins activate immune and endothelial cells and participate in tissue repair. Thus, extracellular S100 proteins may act as chemoattractants, angiogenic factors, and antimicrobials [194]. Despite much progress, little is known about the cell surface receptors that regulate the secretion of S100 proteins. Therefore, the dynamics and regulation of these proteins as well as their functions are not fully understood yet. Data [184] suggest that the functional diversity of these S100 proteins may be regulated by oligomerization and covalent modification, such as S-glutathionylation, cysteinylation, sumoylation, phosphorylation, and intra- and intermolecular disulfide bond formation. Unfortunately, whether post-translational forms of S100 proteins are secreted into the extracellular environment, as these modifications affect the structure and function of these proteins, and consequently the functioning or disorders of the body, has not been largely determined.

Another important aspect is that most of the S100 proteins are encoded in the epidermal differentiation complex (EDC) located on human chromosome 1q21, where a number of chromosomal abnormalities occur linked to epidermal maturation and inflammation [11,19]. This results in an abnormal expression of some S100 genes associated with several human diseases, including psoriasis. Hence, S100 family members may be valuable in the diagnosis of some diseases, as biomarkers of clinical treatment and disease progression and severity [11,12,183,184]. Moreover, S100 proteins are considered potential drug targets for the inhibition of their pathological activities, which may improve therapies. Unfortunately, there are certain gaps in our understanding of the role of S100 proteins in the pathogenesis of several diseases, which provide a wide area for future investigations into this correlation. The research focused on S100 proteins will allow a new look at human diseases.

## 6. S100 Proteins as Biomarkers and Therapeutic Targets

Changes occurring during the disease compared to normal tissues at the gene or protein level are now considered an appropriate way to identify disease markers that may be correlated with disease pathogenesis or response to treatment. Many S100 proteins have been identified in body fluids, which can be used as biomarkers for the detection of a specific disease, where an increased level of their expression indicates a pathological state [195,196,197,198]. Due to the wide range of functions of the S100 proteins, various diseases such as chronic inflammatory diseases are associated with altered levels of specific S100 proteins. The specific S100 protein may be induced under pathological conditions in a cell type that does not express it under normal physiological conditions. For this reason, S100 proteins are being discussed as potential biomarkers for many diseases. Members of the S100 family of proteins intensively studied in the course of psoriasis include S100A4 (Figure A1), S100A7 (Figure A2), S100A8 (Figure A3), S100A9 (Figure A4), S100A12 (Figure A5), S100B (Figure A6) and S100A15, indicating their involvement in the pathogenesis of the disease [1,16,17,18,19,22,24,194,199,200]. However, some studies show no correlation between mRNA expression and plasma levels of the protein, which makes it difficult to assess the possibility of using the S100 protein as a biomarker of psoriasis. For example, the results of the research by Duvetorp et al. [18] showed a significant reduction in the S100A8/A9 skin protein level after NB-UVB treatment, while the serum concentration remained unchanged, thus questioning the function as a psoriasis biomarker. Moreover, conflicting data are found in the literature on the expression of S100A7 in the serum of psoriasis patients under different treatment regimens [1,201,202,203]. The reason for these discrepancies remains unclear, but it has been suggested to be due to different ELISA tests used by the research groups. Both the conflicting results and the still poor understanding of the role of S100 proteins in the development of autoimmune diseases and their potential therapeutic significance provide strong arguments for further research focusing on this issue. The potential role of some S100 proteins as biomarkers and therapeutic targets in selected autoimmune diseases and comorbidities is discussed below. The physicochemical properties of selected S100 proteins are shown in Table A1.

Blood levels of S100A12 are elevated in patients with diabetes [204], coronary artery disease [205], and psoriasis [1], and this protein is also used as a biomarker to detect other inflammatory diseases such as systemic juvenile idiopathic arthritis [206] or acute infectious exacerbations common in cystic fibrosis [196]. Moreover, the intense local expression of S100A12 is indicative of a pro-inflammatory function during airway inflammation in cystic fibrosis, suggesting that this protein may serve as a new target for therapies. The anti-inflammatory effect of methotrexate in patients with inflammatory arthritis was associated with a decrease in serum levels of S100A12, suggesting that S100A12 alone may be a therapeutic target in this disease as well [205,207].

S100A4 has proven to be a valuable biological marker and therapeutic target for many types of cancer. Determination of S100A4 levels in tumor tissues or body fluids can predict the prognosis and metastasis of cancer patients in the early stages. Several molecular targeting strategies have been developed [208] for its protein as inhibition of S100A4 expression may reduce metastasis in vivo. It has been reported [209] that S100A4 is a novel biomarker of glioblastoma stem cells, the increased expression of which contributes to the emergence of a metastatic phenotype. Chow et al. [209] also established that S100A4 is a central node in a molecular network that controls stemness and epithelial-mesenchymal transition in glioblastoma, suggesting S100A4 as a novel candidate therapeutic target. Increased urine levels of S100A4 have been reported in patients with a complex inflammatory autoimmune disease, childhood-onset systemic lupus erythematosus, indicating S100A4 role as a marker for lupus nephritis activity [197]. However, there is still a great clinical need to develop new therapeutic agents that act to modulate the expression and activity of S100A4 [15].

TNF-α, IL-17, and IL-22-induced S100A7 have been found to be abundantly expressed in psoriatic lesions or serum from psoriatic patients as well as in atopic dermatitis skin lesions [2,19,24,210,211]. Moreover, an increased level of S100A7 has been found in the cerebrospinal fluid and brain of Alzheimer’s patients as a function of clinical dementia [212]. This study also confirmed the hypothesis that the promotion of S100A7 expression in the brain can selectively promote α-secretase activity in patients with Alzheimer’s disease (AD), preventing the production of amyloidogenic peptides. S100A7 might be developed as a novel surrogate biomarker of therapeutic efficacy for the treatment of AD. Dysregulation of S100A7 is also associated with the occurrence of many malignancies, including, for example, ovarian cancer [213], cervical cancer [213], and prostate cancer [214]. A new therapeutic strategy is the use of neutralizing monoclonal antibodies against S100A7 in the treatment of cancer [215].

In clinical practice, tests are available that allow the determination of the S100B protein, the presence of which in the serum is a marker of melanoma, useful in the diagnosis, prognosis, and treatment monitoring [216,217,218]. Histochemically, S100B is detected in 100% of neuroblastoma and neurofibroma cells, in 50% of malignant peripheral nerve sheath tumors, and in other neoplastic cells. In histopathology, S100B is an established immunohistochemical marker of choice for malignant melanoma [219]. However, the main clinical importance of S100B in the diagnosis of melanoma is the determination of its concentration in the blood serum [220,221]. S100B is a clinical marker of melanoma progression and metastasis useful in serological monitoring of systemic therapy. The survival time in patients with a lower baseline serum concentration of S100B is significantly longer than in patients with a higher concentration, regardless of the stage of cancer. In the diagnosis of melanoma, the determination of serum S100B is a marker of greater diagnostic utility than classic markers such as lactate dehydrogenase or alkaline phosphatase.

Increased concentration of S100A8/A9 (calprotectin) in serum has been observed in patients with coronary artery disease [222] and may act as a serum biomarker of obesity in patients without type 2 diabetes [223]. Furthermore, calprotectin has also been shown to be a useful biomarker of disease activity in the treatment of inflammatory bowel diseases (IBDs) such as Crohn’s disease [224]. Importantly, the detection of S100A8/A9 in feces can be used to differentiate IBD from irritable bowel syndrome [225]. S100A8/A9 proteins are involved in various types of cancer. Their increased expression indicates a key role in inflammation-related cancers [226], including chronic lymphocytic leukemia [227], breast cancer [228], laryngeal cancer [229], hepatocellular carcinoma [230], and bladder cancer [231]. S100A8/A9 proteins represent promising biomarkers for assessing the risk potential of various types of cancer in molecular pathology. A higher baseline serum calprotectin level may predict that patients with psoriasis vulgaris will experience improvement after treatment with methotrexate [17]. Moreover, a higher concentration of S100A8/A9 was associated with the risk of relapse of the disease after discontinuation of methotrexate. S100A8/A9 may, therefore, be a promising predictive marker of psoriasis severity.

Other members of the S100 family of proteins may prove to be useful biomarkers for future applications, and therapies targeting the S100 protein may prove to be useful possibilities under certain clinical conditions. However, their pathophysiological implications still require further clarification before they can be successfully investigated in a clinical context. Currently, proposed therapeutic strategies targeting S100 proteins include, among others, inhibition of S100 protein expression, targeted degradation, and antibody-mediated binding of S100 proteins [184,232,233]. The most common therapeutic approaches include inhibition of S100 protein expression using microRNA-, small interfering RNA- or short hairpin RNA-based knockdown of S100 proteins using neutralizing antibodies or using specific small-molecule inhibitors. While some inhibitors appear to be effective by inhibiting S100 gene transcription, others inhibit S100 protein activity by disrupting the interaction between S100 proteins and their targets [233]. Target binding cleavages of S100 proteins that are exposed to calcium ion binding can easily bind small molecules [184]. Consequently, considerable success has been achieved in identifying small molecules that block S100–target protein interactions. Several anti-allergy drugs, such as cromolyn, amlexanox, tranilast, and olopatadine, have been reported to bind multiple S100 proteins [234,235,236]. Other anti-allergy drugs have been found to bind to the S100A12 protein, blocking RAGE signaling and subsequent NF-κB activation [237]. Other examples of small-molecule S100 inhibitors include covalent inhibitors that modify Cys residues in helix IV of the S100B and S100A4 proteins. Despite the proximity of these cysteines to the C-terminal EF-hand, their modification does not affect Ca^2+^ binding but disrupts Zn^2+^ mediated conformational rearrangements in S100B and target binding to both S100A4 and S100B [238,239]. Selectivity of this modification is an issue. A covalent inhibitor of S100A4 and S100B, 2,3-bis [2-hydroxyethylsulfanyl]-1,4-naphthoquinone, also inhibits the activity of many protein tyrosine phosphatases by modifying the Cys active site [240,241]. Small-molecule inhibitors that inhibit S100 gene transcription have also been found. Calcimycin, a calcium ionophore, and sulindac sulfide (sulindac), a non-steroidal anti-inflammatory drug, inhibit the expression of β-catenin, leading to reduced levels of target genes, including S100A4 [242,243,244]. Treatment of mice with these inhibitors resulted in reduced tumor growth, reduced invasion, and fewer colorectal cancer metastases at least in part due to lower levels of S100A4 [242,244]. In addition to disrupting the interaction between S100 proteins and their targets, targeting covalent modifications such as S-nitrosylation, S-glutathionylation and phosphorylation may be a promising strategy because these modifications affect the function of S100 proteins [233]. Therefore, targeting these modifications may provide an indirect way to modulate S100 structure or function, thereby affecting pathophysiology and disease progression [15,184]. Function-blocking antibodies targeting receptors and ligands have been widely used as therapeutic agents to treat many pathologies, including immune disorders [245,246,247,248]. Given the extensive evidence indicating that extracellular S100 proteins mediate the inflammatory response in many pathological conditions primarily through cell receptor signaling, the use of antibodies that block S100 function may therefore be an effective therapeutic strategy for treating these conditions. Limiting S100A8/A9 activity with small-molecule inhibitors or neutralizing antibodies has been observed to alleviate pathological conditions in mouse models. Some quinoline-3-carboxamides, compounds currently under investigation for the treatment of human autoimmune and inflammatory diseases, interact with S100A9 and the S100A8/A9 complex, thereby blocking their interaction with TLR4 or RAGE and inhibiting TNF-α release in vivo [249]. Blockade of S100A8/A9 has also recently been observed to reduce inflammatory processes in mouse models of arthritis [250]. Importantly, it has been suggested that S100A8 would be a good target against obesity-induced chronic inflammation [251]. In addition, a monoclonal antibody targeting extracellular S100A7 was designed [215]. It could be shown that this antibody against S100A7, named 6F5, blocks the S100A7/RAGE interaction, thereby inhibiting S100A7-induced MMP9 activity, leading to reduced tumor growth, cell migration and angiogenesis in a xenograft cancer model. Several miRNAs were introduced to target S100 protein expression [233]. Among them, two miRNAs, namely miR-187-3p [252] and miR-149-3p [253], were found to downregulate S100A4 expression. S100A7 expression could be downregulated by miR-26b-5p, leading to reduced proliferation, migration and invasion of intrahepatic cholangiocarcinoma in vitro [254]. Similar effects were observed for miR-24, a miRNA targeting S100A8, which inhibited the proliferation and invasion of laryngeal cancer cells [255]. Elucidating the mechanisms of action of S100 proteins in the pathophysiology of human diseases may lead to the development and application of new, more effective therapeutic approaches. This review describes several promising approaches to using S100 proteins as valuable tools for treating autoimmune disorders. However, much more research is needed to broadly define S100 proteins as reliable biomarkers and to identify and further optimize safe and effective S100 therapies. A better understanding of the role of S100 proteins will greatly benefit new clinical applications.

## 7. Koebnerisin (S100A15)

Gene duplication and variation during primate evolution led to an increase in member number and thereby diversity within the S100 protein family [256]. In 2003, during the analysis of differential gene expression in psoriasis, a new member of the S100 family was discovered, namely S100A15 [20]. S100A15 is a 101 amino acid protein, in which amino acids 13–48 probably form the N-terminal EF-hand motif, while amino acids 50–85 are part of the canonical C-terminal EF-hand (Figure 4) [257]. Selected physicochemical properties of S100A15 are shown in Table 2. It is worth noting that S100A15 has a basic isoelectric point unusual for S100 proteins, which is probably due to the loss of acidic and the introduction of basic amino acids in the acidic C-terminus [20]. Human S100A15 is overexpressed in “koebnerized” psoriatic skin and, thus, the proposed name is koebnerisin [19,21]. Koebnerisin is encoded within the EDC (chromosome 1q21), which has been identified as one of the psoriasis candidate loci (PSORS4) that has been genetically linked to disturbed differentiation and susceptibility to skin inflammation [22]. Unlike other members of S100, the koebnerisin gene reveals an unusual genomic organization [20,22,24,256]. While most S100 proteins encode a single transcript, two alternatively spliced mRNA-isoforms of koebnerisin have been found: S100A15-S (short isoform) and S100A15-L (long isoform). Both S100A15 transcripts share the same coding region but differ in 3′- and 5′-untranslated region (UTR) length (0.5 kb vs. 4.4 kb) and composition [19,20,194].

The S100A15 gene is organized into three exons with exon 1 being not translated and exons 2 and 3 containing the coding region, and two introns with all exon/intron boundaries following the GT-AG rule [20]. Exon 1 occurs only in the S100A15-L transcript, while the S100A15-S transcript starts with an additional sequence enlarging the 5′-UTR of exon 2, which is not a part of the S100A15-L transcript. Both splice variants are differentially expressed in healthy, non-lesional, and lesional psoriatic skin, which indicates regulation by alternate promoters [194]. In the skin, the expression of the S100A15-L mRNA-isoform is more pronounced than the S100A15-S. In healthy skin, koebnerisin is expressed by both differentiated (granular and cornified layer) and non-differentiated (basal layer) keratinocytes of the epidermis [19,194,256]. Additionally, S100A15 can be found in Langerhans cells, melanocytes, and dendritic cells [186]. Within the pilosebaceous unit, koebnerisin is localized both in the internal and external root sheath. In the dermis, S100A15 can be also detected in endothelial tissue, the basal layer of the sebaceous gland as well as in smooth muscle cells. Koebnerisin is secreted into the extracellular space, where it acts as an AMP against *Escherichia coli, Staphylococcus aureus, and Pseudomonas Aeroginosa* [258]. In the model of psoriasis, the inflammation susceptibility of keratinocytes is mediated by the upregulation of S100A15 [22]. When secreted to the extracellular medium, koebnerisin drives the production of some proinflammatory cytokines (autocrine loop) and attracts immunocytes (paracrine effect). Then, it establishes a subtle inflammation-prone environment.

An external stimulus such as trauma leads to Koebnerization and the further proinflammatory cascade is amplified (inflammation priming). During inflammation, levels of S100A15 significantly increase and this protein is now distributed to the whole epidermis, which might be due to the altered calcium gradient along with disturbed maturation in the psoriatic skin [194]. In addition to the regulation of koebnerisin by disturbed calcium-induced epidermal differentiation, also proinflammatory environment contributes to the expression of S100A15 in human skin. Psoriasis is characterized by increased epidermal proliferation, abnormal keratinocytes differentiation, and infiltrating inflammatory cells [192]. In inflamed skin, the levels of immune cells such as neutrophils, lymphocytes, granulocytes, and macrophages increase significantly. The infiltrating cells express high amounts of proinflammatory cytokines, such as IFNγ, IL-12, IL-23, and IL-17 [23]. During chronic inflammation, koebnerisin is mainly induced by Th1-, Th17- and Th22-derived cytokines, such as IL-17A, TNF-α, IL-22, IL-1β, and IFN-γ that create a characteristic psoriatic proinflammatory milieu [19,23,192,194]. Data [258] suggest that IL-17A is the principal inducer of S100A15 in human keratinocytes. The cytokine environment with the increase in S100A15 expression catalyzes the vicious cycle of inflammation. Koebnerisin primes keratinocytes to enhance the production and secretion of subsequent immunotropic cytokines such as IL-6, IL-8, and TNF-α, which are crucial in the development of psoriatic lesions [23,192]. Inflammation priming and amplification require S100A15 binding to the as-yet-unknown pertussis toxin-sensitive Gi-protein-coupled receptor (GiPCR) [19,22,23]. Koebnerisin potentiates inflammation by acting directly as a chemoattractant for leukocyte subsets, especially granulocytes and monocytes, further increasing the number of inflammatory cells infiltrating the skin and amplifying a pro-inflammatory feedback loop in psoriasis. While koebnerisin induces a proinflammatory response by itself, its proinflammatory activity is potentiated when it acts with highly homologous S100A7 (psoriasin) which indicates that they contribute to inflammation independently, but synergistically [200,256]. Inflammatory products of psoriatic skin are then released into the systemic circulation reflecting the severity and extent of psoriatic lesions [260]. As a result, it potentiates systemic inflammation and contributes to the development of psoriatic comorbidities [24,260]. Moreover, Batycka-Baran et al. [24] confirmed the expression of S100A15 by human leukocytes, where the S100A15 gene was also transcribed into two alternate splice variants. In contrast to the skin, S100A15-S was a more prominent isoform in all investigated leukocyte subsets, but both transcripts were predominantly expressed in lymphocytes and monocytes. Then, koebnerisin acted as a danger molecule (‘alarmin’) by priming these immune cells to produce proinflammatory cytokines. Compared to the healthy control, there were increased levels of koebnerisin transcripts in peripheral blood mononuclear cells of psoriatic patients. Therefore, circulating leukocytes may contribute to the increased level of S100A15 both in skin lesions and in the serum of patients with psoriasis, which indicates that extracellular koebnerisin, may be considered a biomarker associated with this disease. This is also confirmed by the fact that the use of narrow-band UVB therapy significantly reduces the amount of koebnerisin expressed in the leukocytes of patients with psoriasis as well as calcipotriol contributes to a decreased level of S100A15 in psoriatic skin, which gives this protein a valuable role as a marker of therapeutic response in psoriasis [24,192]. Researches [24,260,261] show a broader spectrum of S100A15 activity as a mediator of inflammation, not only in the skin but also beyond, which may have systemic consequences in the form of comorbidities in psoriasis, such as atherosclerosis and obesity. Awad et al. [260] evaluated the role of serum koebnerisin as a potential link between psoriasis and atherosclerosis. The results of these studies confirmed the usefulness of S100A15 as a marker of subclinical atherosclerosis in patients with psoriasis. Thus, targeting the S100A15-mediated loop may be an excellent approach in the treatment of psoriasis and comorbidities in the future.

## 8. The Function of Koebnerisin in Other Immune-Mediated Inflammatory Diseases

In the last decade, some researchers have addressed other disease fields of SA10015. Batycka-Baran et al. [262] showed a relevant role of koebnerisin as a protein that might exert a proinflammatory effect in rosacea. The level of S100A15 was upregulated within the epidermis and dermis in rosacea lesional skin compared with the healthy control, suggesting its role in the regulation of keratinocytes’ functions. It is well known that koebnerisin can prime leukocytes and keratinocytes to enhance the production of proinflammatory cytokines such as TNF-α and IL-1β with a significant role in rosacea. The study showed in turn that TNF-α enhanced the expression of S100A15 in keratinocytes and fibroblasts, which might create a vicious circular cycle of inflammation. Moreover, it was found that koebnerisin may exert additional proangiogenic effects by stimulating keratinocytes and fibroblasts to increase the production of the potent proangiogenic mediator vascular endothelial growth factor (VEGF). In addition, S100A15 can trigger fibroblasts to increase the expression of matrix metalloproteinase 9 (MMP-9), which has a damaging effect on skin components, thereby stimulating innate immune responses and inflammatory processes. In conclusion, koebnerisin may be a new player in the pathogenesis of rosacea. Balancing the activity of certain antimicrobial proteins may be a target for future therapeutic interventions in rosacea.

Another study by Batycka-Baran et al. [263] showed increased expression of koebnerisin in lesional and perilesional skin in patients suffering from hidradenitis suppurativa compared to a healthy control. Similar results were also obtained by Zouboulis et al. [264], who detected a strong overexpression of S100A15 in hidradenitis suppurativa skin, especially in *stratum granulosum*. Real-time PCR and immunofluorescence analysis with specific monoclonal antibodies were used to examine the expression of S100A15 in the skin [263]. Patients with hidradenitis suppurativa had not received any local or systemic anti-inflammatory therapy for at least 8 weeks before the study. Disease severity was graded according to Hurley classification (stage I-4, stage II-5, and stage III-5). Two punch biopsies were taken from each volunteer, one of the inflammatory lesions and the other 2 cm from the affected skin. S100A15 was found to be overexpressed in the epidermal suprabasal and basal compartments of the perilesional and lesional skin in patients with hidradenitis suppurativa. It has been suggested that the overexpression of koebnerisin in the perilesional hidradenitis suppurativa skin may indicate its role in the early phase of disease pathogenesis, as well as contribute to the hidradenitis suppurativa susceptibility. Skin biopsies from clinically uninvolved, perilesional hidradenitis suppurativa skin show perifollicular and perivascular inflammatory infiltrates. Koebnerisin triggers keratinocytes to increase the production of pro-inflammatory cytokines, including TNF-α, IL-6, and IL-8. Elevated levels of these early induced proinflammatory mediators were found in hidradenitis suppurativa skin. Furthermore, S100A15 is involved in the pathogenesis of hidradenitis suppurativa as a chemoattractant for neutrophils and monocytes/macrophages. Therefore, S100A15 can promote the development of inflammatory processes in the skin. The same study showed an increase in S100A8 levels in lesional skin compared to skin in the lesion area, but, in contrast to the S100A15, there was no significant difference in S100A8 expression in the perilesional hidradenitis suppurativa skin compared to healthy skin. Nevertheless, S100A8 may enhance inflammation in patients with hidradenitis suppurativa by stimulating keratinocytes to increase the production of proinflammatory mediators and attracting other immune cells. In a later study, Batycka-Baran et al. [265] investigated S100A15 and S100A4 serum levels in patients suffering from hidradenitis suppurativa and their association with disease severity, C-reactive protein (CRP) serum levels, and other demographic and clinical data. The S100A4 protein showed a statistically significant increase in concentration in plasma samples of hidradenitis suppurativa subjects compared to controls. Significant differences in S100A4 levels were also observed between different Hurley stages with the highest concentration in patients in Hurley II stage. The S100A4 serum concentration in the patients in Hurley stage II and III was significantly elevated as compared to those of the healthy controls and subjects with hidradenitis suppurativa in Hurley stage I, but there was no significant difference in the S100A4 serum levels between the patients in Hurley stages II and III. They also found no significant correlations between serum S100A4 concentration and CRP levels, body mass index (BMI), smoking, and other demographic and clinical data. In contrast to S100A4 protein, the study showed no significant differences (*p* > 0.05) in serum S100A15 levels between whole individuals with hidradenitis suppurativa (156.1 ± 133.8 pg/mL) and healthy volunteers (153.9 ± 134.8 pg/mL). However, an association has been shown between disease severity (estimated based on Hurley staging system) and koebnerisin serum levels in patients with hidradenitis suppurativa (50.8 ± 30.9 pg/mL, 151.5 ± 115.7 pg/mL, 317.1 ± 101.0 pg/mL in Hurley stage I, II and III, respectively). There were statistically significant differences in concentration of koebnerisin in serum between the Hurley stages (*p* < 0.0001), as well as between S100A15 serum levels in patients in Hurley stage III compared with controls (*p* = 0.0013). Moreover, a positive correlation was found between the serum concentration of S100A15 and CRP in patients with hidradenitis suppurativa, which may suggest an association between the disease and increased cardiovascular risk resulting from chronic systemic inflammation. However, as with S100A4, no relationship was found between S100A15 levels and BMI, smoking, or other demographic or clinical data. The authors proposed S100A4 and S100A15 as novel serum biomarkers for monitoring hidradenitis suppurativa progression and suggested their role in the pathogenesis of the disease by promoting inflammation and fibrosis.

The genes of the S100 protein family are dysregulated during carcinogenesis, and today, some S100 members have been established as markers of tumor progression [266,267,268,269]. Psoriasin and koebnerisin form the highly homologous S100 subfamily and are regulated throughout tumor progression in epithelial cancers [266]. Despite their 93% of sequence homology, S100A7 and S100A15 differ in expression, function, and mechanism of action and are, therefore, exemplary of the diversity within the S100 family. The corresponding single ortholog mS100a7a15 in mice shares the expression and functional properties of the two human proteins. Psoriasin and koebnerisin are co-expressed in mature epithelial cells of mammary lobules and ducts, as well as in differentiated epithelial skin layers, but are difficult to distinguish. Their expression is induced along with markers of late differentiation in calcium-differentiated keratinocytes. In epithelial carcinomas, S100A7 and possibly S100A15 are often elevated at early tumor stages, such as in pre-invasive carcinomas. Within tumor tissue, both proteins are expressed in well-differentiated tumor cells and show an expression pattern like that observed in normal mature epithelial cells. S100A7 is downregulated in adjacent invasive cancer tissues; however, once the expression persists, the nuclear translocation of psoriasin is related to a poor clinical prognosis. Within the nucleus, psoriasin is assumed to bind and trigger c-jun activation domain-binding protein-1 (Jab1). Nuclear translocation of psoriasin and Jab1-dependent effects leads to increased cell survival and proliferation. In invasive breast carcinomas, S100A7 and S100A15 are co-regulated and related to estrogen/progesterone receptor-negative and more aggressive tumors. Koebnerisin nuclear translocation has not been studied, but because of mutations within the Jab1 binding site, it may not be able to bind to Jab1 compared to psoriasin. A similar suggestion was made by Wolf et al. [270]. Oncogenic effects may also be mediated by the release of the S100 protein into the extracellular space where it interacts with cell surface receptors, for example, RAGE, which is responsible for maintaining inflammation and promoting carcinogenesis [266]. Increased serum levels of S100A7 were considered a potential marker of epithelial cancer progression. Extracellular psoriasin can bind to RAGE and thus activate NF-κB, which controls the transactivation of several genes involved in immune responses as well as cell proliferation and apoptosis. In inflammation-related carcinogenesis, NF-κB is a key player in helping precancerous and malignant cells escape from tumor surveillance mechanisms by activating anti-apoptotic genes. Koebnerisin is unable to induce RAGE signaling but interacts with an as-yet-unknown Gi protein-coupled receptor. S100A7 and S100A15 are two proteins that are exploited by the tumor not only to adapt cellular signaling pathways that regulate the survival of the tumor but also to modify the surrounding microenvironment to escape tumor surveillance, as well as to promote cancer cell migration. In a breast cancer model, the mouse S100a7a15 ortholog induced matrix metalloproteinases (MMP-2) and angiogenic factors, such as VEGF resulting in enhancing tumor malignancy. Moreover, mS100a7a15 can recruit leukocytes and tumor-associated macrophages (TAM) via RAGE/Stat3 signaling and thus promote tumor progression and metastasis. The human orthologs are chemoattractants and able to recruit myeloid cells, such as monocytes, but their participation in the evasion of tumor surveillance by attracting TAM has not been investigated yet. Compared to S100A7, koebnerisin is produced additionally by tumor surrounding non-epithelial cells such as dendritic cells, epithelial-derived myoepithelial cells around acini, and surrounding blood vessels. Psoriasin and koebnerisin can mediate the immune response by stimulating the secretion of TNF-α, IL-1, IL-6 and IL-8. In immune cells, these pro-inflammatory cytokines lead to the NF-κB-dependent secretion of growth factors that enhance the proliferation and survival of cancer cells. In addition, the attracted macrophages, mast cells, and neutrophils may also enhance non-specific immune responses that may lead to enhanced tumor growth. Despite the few reports about the opposing effects of psoriasin and koebnerisin in multifunctional pathways and in mechanisms that are known to affect epithelial carcinogenesis, still, little is known about the recently discovered koebnerisin. Several functions that are referred to psoriasin might actually be due to koebnerisin signaling. Their different properties are very important reasons for the need to discriminate psoriasin and koebnerisin in epithelial homeostasis, inflammation, and carcinogenesis.

The results obtained by Yung-Che et al. [269] provide evidence that koebnerisin enhances metastasis in lung adenocarcinoma in vivo and in vitro by hypomethylation of DNA in the gene promoter region and downstream mediators focused on CTNNB1. Researchers found that hypomethylation of the S100A15 promoter at three CpG sites and its increased expression was associated with both a higher metastatic potential and poorer outcomes in patients with pulmonary adenocarcinoma. This phenomenon has been verified in lung adenocarcinoma cell lines with high and low metastatic properties. As the accumulation of koebnerisin in the nucleus has been demonstrated by immunochemical staining in patients with distant metastatic pulmonary adenocarcinoma, it is speculated that its nuclear translocation from under the cell membrane is the first step to exerting its further oncogenic effects. More research is needed to elucidate the relationship between S100A15 promoter DNA hypomethylation and its nuclear translocation. The study did not reveal a clinical relationship with S100A15 in the other two pathological types of lung cancer, namely squamous cell carcinoma and small-cell carcinoma. This could be for several reasons. First, the interaction between the epidermal growth factor receptor and the S100A family can promote angiogenesis and metastasis in various cancers, while the proportion of EGFR mutations is relatively small in these two types of lung cancers. Second, some members of the S100A family contribute to the progression of squamous cell carcinoma, while others maintain the differential status of the epithelium and contribute to a less invasive type of cancer. Although nuclear S100A15 has been relatively strongly expressed in squamous cell carcinoma, its biological function in this type of lung cancer remains to be established. Third, little expression of the S100A family is found in various small-cell neoplasms. Koebnerisin may not play a key role in small-cell lung cancer. On the other hand, downstream S100A15 signaling, which may be important for cancer cell survival, remains largely unknown. Next-generation sequencing data identified 518 differentially expressed genes upregulated by S100A15 and 1378 differentially expressed genes downregulated by S100A15, the former having been mapped to 46 seed genes of the subnetwork. Among them, CTNNB1, ZEB1, CDC42, HSP90AA1, BST2, PCNA, and E2F1 have been shown to promote lung cancer progression, while SAMHD1, HRAS, and NQO1 serve as tumor suppressor genes. Thus, S100A15 promotes tumor progression in adenocarcinoma of the lung. S100A15 can exert its oncogenic function, initiated by DNA hypomethylation in the promoter region of the gene and mediated by downstream genes focused on CTNNB1. Both increased expression of S100A15 and hypomethylation of the promoter DNA of its gene can serve as biomarkers predicting high metastatic potential and poor outcomes in adenocarcinoma patients. Further research into the functions of S100A15 and its epigenetic regulation could provide a potential treatment strategy for lung cancer.

## 9. Conclusions

Psoriasis is the most diagnosed dermatosis in clinical practice with unknown etiology. One of the characteristic abnormalities in psoriasis is the excessive production of antimicrobial peptides and proteins, which play a key role in the innate immune system and host defense against pathogens. AMPs are currently being considered for the treatment of inflammatory skin diseases and infected-wound-healing improvement. Targeted therapy strategies are being developed, and early results indicate that AMPs affect processes in diseased skin. Nevertheless, only a small fraction of the AMPs reported so far have been able to successfully complete all phases of clinical trials and become available to patients. This is mainly due to low bioavailability and metabolic stability resulting from sensitivity to proteolytic degradation as well as the costly production of AMPs. S100 protein biochemistry continues to provide a rich area of new investigations and the evaluation of the physical functions of these proteins will provide new insight into human immune-mediated diseases. Determination of changes in the concentration of S100 proteins in biological samples at different physiological states could help reflect their use as diagnostic biomarkers in the future. Understanding the mechanisms of action of S100 proteins in the pathophysiology of immune-mediated inflammatory diseases may also lead to the development and application of new, more effective therapeutic methods. Future research should, therefore, focus on the validation of S100 proteins as biomarkers for early disease detection and prognosis and the development of new strategies based on anti-S100 therapies or targeting the S100–receptor axis. The current, promising therapeutic approach in psoriasis focuses on the inhibition of pro-inflammatory pathways, crucial in the pathogenesis of this disorder. Koebnerisin is involved in the early steps of inflammation in psoriasis, so inhibition of the S100A15-mediated inflammatory cascade might constitute an innovative therapeutic option. The great importance of S100A15 protein and, at the same time, the little knowledge about its effect on skin pathophysiology as well as the development of comorbidities are strong arguments for the need for further research focusing on this issue.

## Figures and Tables

**Figure 1 molecules-27-06640-f001:**
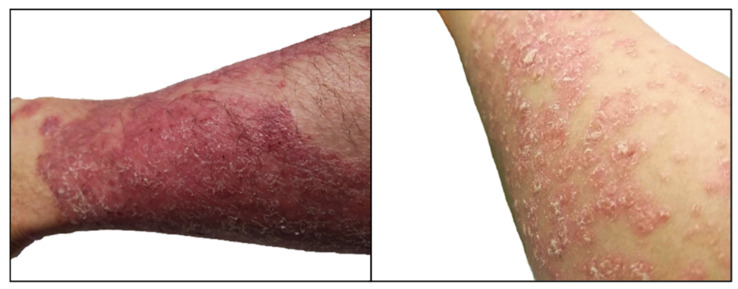
Clinical manifestations of plaque psoriasis on the hand from two different patients.

**Figure 2 molecules-27-06640-f002:**
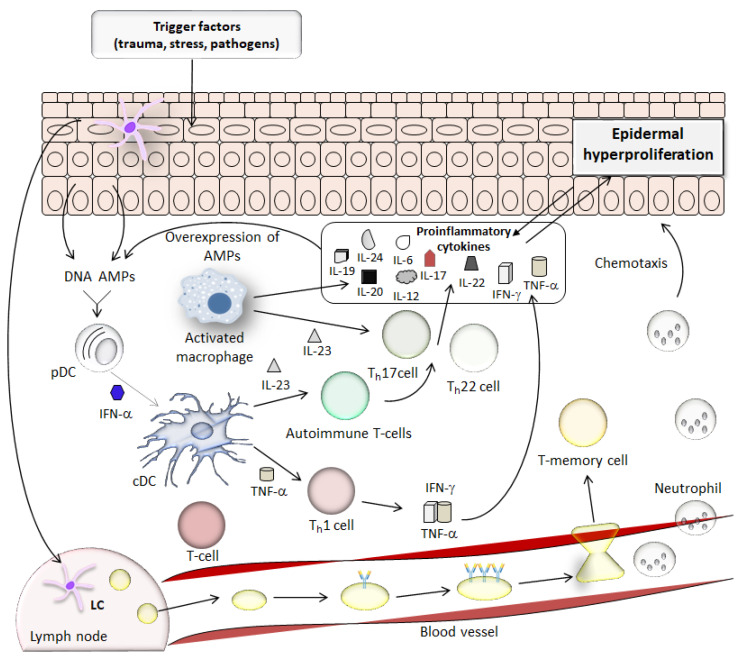
Schematic representation of psoriasis pathogenesis. AMPs—antimicrobial peptides and proteins, pDC—plasmacytoid dendritic cell, cDC—conventional dendritic cell, Th cell—T helper cell, IL—interleukin, TNF—tumor necrosis factor, IFN—interferon, LC—Langerhans cell.

**Figure 3 molecules-27-06640-f003:**
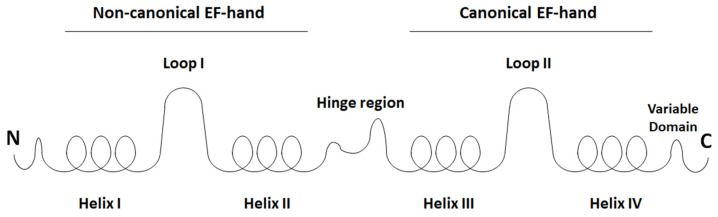
Schematic representation of the S100 protein secondary structure. Each calcium-binding loop (Loop I and Loop II) is flanked by α-helices (Helix I and II for Loop I and Helix III and IV for Loop II). The non-canonical N-terminal EF-hand has approximately 100-fold lower affinity to bind calcium than the canonical C-terminal EF-hand. The hinge region links Helix II with Helix III. C-terminal EF-hand is followed by a C-terminal extension, which is variable among S100 proteins and together with the hinge region have a crucial role in the interaction of each S100 protein family member with specific target proteins.

**Figure 4 molecules-27-06640-f004:**
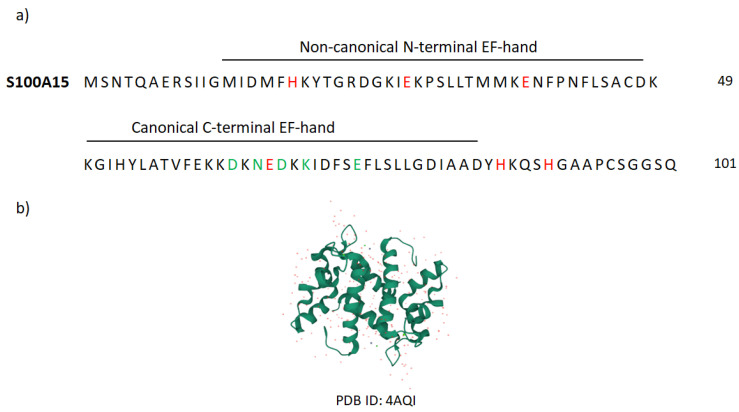
Structures of S100A15 protein: (**a**) predicted amino acid sequence of the human S100A15 (the red color marks those amino acids that bind Zn^2+^, while the green color marks the amino acids that bind Ca^2+^); (**b**) structure of human S100A15 coordinating Zn^2+^ and Ca^2+^—image from the RCSB PDB (rcsb.org) of PDB ID 4AQI [187].

**Table 1 molecules-27-06640-t001:** List of AMP in clinical trials and Food and Drug Administration (FDA) approved AMP for various human infections and diseases [5,6,42].

AMP	Source	Medical Use	Formulation/Route of Administration	FDA Approval/Stage of Development	Company
Oritavancin	Semisynthetic glycopeptide	Acute bacterial skin and skin structure infections	Intravenous	2014	Orbactiv
Multiferon	Leukocyte fraction of human blood	Metastatic renal cell carcinoma	Subcutaneous	2006	Intron/Roferon-A/Roche
Boceprevir	Synthetic peptide	Hepatitis C virus genotype 1	Oral	2011	Victrelis/Merck
Daptomycin (lipopeptide)	*Streptomyces roseosporus*	Complicated skin infections caused by susceptible strains of Gram-positive microorganisms	Intravenous injection	2003	Cubist Pharmaceuticals LLC (Merck & Co.)
PAC-113	Human histatin 3	Oral candidiasis in HIV seropositive patients	Mouth rinse	Phase II	Pacgen Biopharmaceuticals Corporation/Quintiles, Inc.
DPK-060	Human kininogen	Bacterial infections in atopic dermatitis and acute external otitis	Ointment for local application	Phase II	DermaGen AB/Pergamum AB
LL-37	Human (derived from hCAP18)	Hard-to-heal venous leg ulcers	Polyvinyl alcohol-based solution for administration in the wound bed	Phase II	Lipopeptide AB
DiaPep277	Human Heat Shock Protein 60 (Hsp60)	Type 1 Diabetes mellitus	Subcutaneously	Phase III	Andromeda Biotech Ltd.
Omiganan	Bovine indolicidin	Catheter infections, atopic dermatitis, genital warts, acne vulgaris, and rosacea	Topical gel	Phase III/Phase II	Cutanea Life Sciences/Mallinacckrodt
PM060184	*Lithop locamialithistoides*	Cancer	Intravenous	Phase I	PharmaMar (Colmenar Viejo, Madrid, Spain)
Pexiganan	Analog of magainin from African clawed frog	Infected diabetic foot ulcers	Topical cream	Phase III	Dipexium Pharmaceuticals (has merged with PLx Pharma)

**Table 2 molecules-27-06640-t002:** Chemical and physical properties of S100A15 [20,258,259].

Property	Value
Chemical formula	C_497_H_784_N_130_O_153_S_7_
Molecular weight [Da]	11304.902
Net charge	4
Boman Index [kcal/mol]	1.83
Hydrophobic residue [%]	34
Isoelectric point	7.57
Instability index	41.85
Aliphatic index	63.86
GRAVY	−0.572

## Data Availability

Not applicable.

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
