# Peer review of "S100 Proteins as Novel Therapeutic Targets in Psoriasis and Other Autoimmune Diseases"

_molecules, 2022, doi:10.3390/molecules27196640_

Round 1

Reviewer 1 Report

The paper introduces psoriasis and the pathogenesis of the disease. Then, this review mainly describes the involvement of antimicrobial peptides and proteins in inflammatory diseases’ development and therapy. The discussion focuses on S100 proteins, especially koebnerisin, which may be involved in the mechanism in psoriasis, and other immune-mediated inflammatory diseases. 

The paper is overall well, but there are missing details that could make the paper stronger. The following comments can help improve this point. 

1. The figure 2, 3, and 4 are not very clear to read. Please make the texts clear, and increase image resolution. 

2. In Figure 4, the author shows the structure of S100A15 protein, please also give the corresponding PDB ID. Furthermore, also point out the non-canonical N-terminal EF-hand and canonical C-terminal EF-hand in the structure.

3. Although authors highlighted Koebnerisin (S100A15), it is also interesting to show the structures (like Figure 4) and properties (like Table 2) of other members in S100 protein family. 

Author Response

Dear Reviewer,

thank you very much for your comments on our manuscript. The changes we have made in the revision are marked up using the “Track Changes” function.

Comment: The figure 2, 3, and 4 are not very clear to read. Please make the texts clear and increase image resolution.

Author response: The text in the figures has been appropriately enlarged to increase readability. The resolution of the images has been increased by pasting them as bitmaps.

Comment: In Figure 4, the author shows the structure of S100A15 protein, please also give the corresponding PDB ID. Furthermore, also point out the non-canonical N-terminal EF-hand and canonical C-terminal EF-hand in the structure.

Author response: We added the PDB ID of S100A15 protein’s structure (4AQI) in the figure and caption. Moreover, we corrected the EF-hand motifs according to the UniProt database: the 1st EF-hand of S100A15 is at amino acids 13-48 and the 2nd EF-hand of S100A15 is at amino acids 50-85. We have also indicated in red those amino acids that bind zinc ions (Zn2+) and in green the amino acids that bind calcium ions (Ca2+).

Comment: Although authors highlighted Koebnerisin (S100A15), it is also interesting to show the structures (like Figure 4) and properties (like Table 2) of other members in S100 protein family.

Author response: Regarding the commentary, we added the structures of six S100 proteins (similar to figure 4) involved in the pathogenesis of psoriasis as well as other autoimmune diseases. These include: S100A4, S100A7, S100A8, S100A9, S100A12, S100B. A PDB ID was added to each model, and against each sequence the amino acids that make up the EF-hand motifs are marked with lines. It is also shown which amino acids bind calcium and iron ions. To not disrupt the flow of the main text, we added figures of S100 proteins' structures in the form of Appendix A. We also showed chemical and physical properties of selected S100 proteins computed using ProtParam and Prot pi tools as Appendix B (Table B1). The computed parameters include the molecular weight, theoretical pI, atomic composition, instability index, aliphatic index, net charge at pH 7.4, and grand average of hydropathicity (GRAVY).

Reviewer 2 Report

In this manuscript, the authors discussed the S100 proteins as major players in many human diseases with special emphasis on their roles in the pathogenesis of psoriasis and other autoimmune diseases. Although this manuscript presents the current advances of this field, especially on the therapeutic targeting of S100 proteins as potential strategy to ameliorate psoriasis and other autoimmune diseases, there are some writing weaknesses that should be addressed prior to publication.

1. Section 2 of the manuscript is on the “Pathogenesis of psoriasis.” However, after this section, the authors discuss about antimicrobial properties of AMPs, which appears off-tangent from the main topic of the manuscript. In particular, the “Antimicrobial peptides and proteins” section and certain areas of the “Therapeutic potential of AMPs” can be rewritten in a more concise manner and can then be incorporated in the ‘Introduction.”

2. L 241:  “…against many pathogenic microorganisms and (have) anti-inflammatory properties…”

Provide examples of AMPs that have anti-inflammatory properties and cite references where they were proposed or demonstrated to be promising in ameliorating psoriatic pathogenesis (and other autoimmune diseases). In contrast, the authors should also discuss that a subset of the AMP family of proteins have pro-inflammatory effects.

3. L 297-298: “…applications of AMPs in clinical trials are mainly limited to topical applications…”

The authors should include in this review that certain AMPs are administered by intravenous in some clinical trials. These AMPs include the p2TA(AB103), Neuprex(rBPI21), Ghrelin, and hLF1-11.

4. Include in the discussion the strategies to improve the stability (e.g., resistance from proteases) and/or decrease toxicity (e.g., prevent non-specific interactions with the host’s cell membranes) of certain AMPs especially by chemical modification.

5. L 467-481

This whole paragraph starts with S100A4 as biomarker for glioblastoma cancer cells, then certain lupus types, and then back to many types of cancer – this requires rewriting into a more organized flow of thought.

6. A huge chunk of the manuscript is on S100 proteins as biomarkers for many cancer types, cardiovascular diseases, and obesity. These can affect the coherence of the entire manuscript – which is supposedly on S100 proteins in psoriasis and other autoimmune diseases.

7. L 532: “…amino acids 12-39 probably form the N-terminal EF-hand motif…”

In UniProt, the 1st EF-hand of S100A15 is at amino acids 13-48.

L 532-533: “…while amino acids 54-82 are part of the canonical C-terminal EF-hand…”

In UniProt, the 2nd EF-hand of S100A15 is at amino acids 50-85.

8. L 547: “…human S100A15 bound to zinc and calcium…”

This can be rewritten as “human S100A15 coordinating Zn2+ and Ca2+.” Indicate which amino acids (in the S100A15 amino acid sequence and cartoon structure) bind these ions.

9. The manuscript lacks some mechanistic details (in the molecular level) on the role of S100 proteins in psoriasis and other autoimmune diseases.

10. The manuscript lacks proposed therapeutic strategies and approaches in targeting the S100 proteins (e.g., inhibition of S100 protein expression, targeted degradation, antibody-mediated binding of S100 proteins, etc.).

Author Response

Dear Reviewer,

thank you very much for the valuable suggestions and comments. Your critical comments help us to improve our paper writing skills. We have answered the questions one by one, and have made careful modifications to the manuscript. The changes we have made in the revision of the manuscript are marked up using the “Track Changes” function.

Comment: Section 2 of the manuscript is on the “Pathogenesis of psoriasis.” However, after this section, the authors discuss about antimicrobial properties of AMPs, which appears off-tangent from the main topic of the manuscript. In particular, the “Antimicrobial peptides and proteins” section and certain areas of the “Therapeutic potential of AMPs” can be rewritten in a more concise manner and can then be incorporated in the ‘Introduction.”

Author response: With reference to this commentary, the separation of a separate subsection on antimicrobial peptides and proteins is intended to introduce the reader outside the discipline to the basic properties and functions of AMPs, which are crucial when considering these compounds as potential drug candidates. The antimicrobial properties of AMPs form the basis of their major applications in clinical trials, so we feel it is reasonable to briefly (one paragraph) introduce this aspect at the beginning of the subsection on AMPs. It should also be noted that the antimicrobial activity of AMPs is not insignificant in their involvement in the course of autoimmune diseases, which can be illustrated by the example of atopic dermatitis, in which reduced levels of certain AMPs are observed, and consequently patients are more susceptible to S. Aureus infections. 

Comment: L 241:  “…against many pathogenic microorganisms and (have) anti-inflammatory properties…”

Provide examples of AMPs that have anti-inflammatory properties and cite references where they were proposed or demonstrated to be promising in ameliorating psoriatic pathogenesis (and other autoimmune diseases). In contrast, the authors should also discuss that a subset of the AMP family of proteins have pro-inflammatory effects.

Author response: We have given examples of AMPs that have anti-inflammatory properties and cited references where they have been proposed or shown to be promising therapeutic targets in psoriasis or other autoimmune diseases. We discussed the dual role of AMPs in autoimmunity, taking into account that they exhibit pro-inflammatory effects in addition to anti-inflammatory effects, citing LL-37 in selected autoimmune diseases as a prime example.

Comment: L 297-298: “…applications of AMPs in clinical trials are mainly limited to topical applications…”

The authors should include in this review that certain AMPs are administered by intravenous in some clinical trials. These AMPs include the p2TA(AB103), Neuprex(rBPI21), Ghrelin, and hLF1-11.

Author response: We have included in this review that some AMPs are administered intravenously in some clinical trials. We described proposed AMPs such as p2TA (AB103), Neuprex (rBPI21), Grelin, and hLF1-11.

Comment: Include in the discussion the strategies to improve the stability (e.g., resistance from proteases) and/or decrease toxicity (e.g., prevent non-specific interactions with the host’s cell membranes) of certain AMPs especially by chemical modification.

Author response: We included in the discussion the strategies to increase the stability and effectiveness of AMPs and reduce their cytotoxicity and non-targeted side effects. Strategies that are discussed at length in this review include: cyclization, incorporation of atypical amino acids into the AMP sequence, side/end chain modifications, peptide conjugation, nanoparticle formulations, or peptidomimetics.

Comment: L 467-481 This whole paragraph starts with S100A4 as a biomarker for glioblastoma cancer cells, then certain lupus types, and then back to many types of cancer – this requires rewriting into a more organized flow of thought.

Author response: We rewrote the paragraph into a more structured flow of thought.

Comment: A huge chunk of the manuscript is on S100 proteins as biomarkers for many cancer types, cardiovascular diseases, and obesity. These can affect the coherence of the entire manuscript – which is supposedly on S100 proteins in psoriasis and other autoimmune diseases.

Author response: The link between autoimmunity and cancer is well established. Autoimmune diseases are often associated with malignant neoplasms, and some malignant diseases are also associated with an increased risk of developing autoimmune disorders. Cancer has been linked to certain autoimmune disorders (AIDs) such as scleroderma and myositis. On the other hand, many autoimmune disorders and immunosuppressive therapies have been associated with an increased risk of cancer. Obesity and cardiovascular diseases, in turn, are the main diseases accompanying autoimmune diseases, so S100 proteins can not only serve as markers of specific diseases resulting from disorders of the immune system but also help in the diagnosis of comorbidities, such as obesity in psoriasis. All the disease entities described by us are therefore closely related and fit into the main concept of this review.

Comment: L 532: “…amino acids 12-39 probably form the N-terminal EF-hand motif…”

In UniProt, the 1st EF-hand of S100A15 is at amino acids 13-48. L 532-533: “…while amino acids 54-82 are part of the canonical C-terminal EF-hand…” In UniProt, the 2nd EF-hand of S100A15 is at amino acids 50-85.

Author response: Corrected in the text and in point a) of figure 4.

Comment: L 547: “…human S100A15 bound to zinc and calcium…”

This can be rewritten as “human S100A15 coordinating Zn2+ and Ca2+.” Indicate which amino acids (in the S100A15 amino acid sequence and cartoon structure) bind these ions.

Author response: We have indicated in red the amino acids that bind Zn2+ and in green the amino acids that bind Ca2+ in the amino acid sequence. We also rewrote the caption as proposed and added the PDB ID.

Comment: The manuscript lacks some mechanistic details (in the molecular level) on the role of S100 proteins in psoriasis and other autoimmune diseases.

Author response: Because of the breadth of the article and the issues it raises, the mechanistic details of the role of S100 proteins in psoriasis and other autoimmune diseases were not the subject of this review. We agree that this is an important point, but could be the subject of another manuscript.

Comment: The manuscript lacks proposed therapeutic strategies and approaches in targeting the S100 proteins (e.g., inhibition of S100 protein expression, targeted degradation, antibody-mediated binding of S100 proteins, etc.).

Author response: We added some therapeutic strategies to targeting the S100 proteins.

Reviewer 3 Report

The review article entitled "S100 proteins as novel therapeutic targets in psoriasis and other autoimmune diseases" is very interesting, however, similar articles are already online. Therefore, authors need to explain novelty of their review. 

Also, authors need to explain the role of Vitamin D, Melatonin in S100 Proteins expression and regulation.

Author Response

Dear Reviewer,

thank you very much for your opinion on our manuscript entitled ‘S100 proteins as novel therapeutic targets in psoriasis and other autoimmune diseases’. The changes we have made in the revision are marked up using the “Track Changes” function.

Comment: Therefore, authors need to explain novelty of their review.

Author response: Our review offers some new information and makes the case for essential advances that were carried out in the context of other pathogenesis of diseases, in which S100A15 can be included and which was studied in the last 5 years. We prepared the paper with a wider scope that covers all the literature of importance, makes the case, and justifies the need for the review. We discussed the S100 proteins as major players in many human diseases with special emphasis on their roles in the pathogenesis of psoriasis and other autoimmune diseases as well as comorbidities. This manuscript presents the current advances in this field, especially on the therapeutic targeting of S100 proteins as a potential strategy to ameliorate autoimmune diseases. We also broadly discussed the strategies to improve the stability and/or decrease the toxicity of certain AMPs, especially by chemical modification, as well as proposed therapeutic strategies and approaches in targeting the S100 proteins.

Comment: Also, authors need to explain the role of Vitamin D, Melatonin in S100 Proteins expression and regulation.

Author response: As recommended, we explained briefly the role of melatonin and vitamin D in the expression of S100 proteins.

Round 2

Reviewer 3 Report

The manuscript entitled "S100 proteins as novel therapeutic targets in psoriasis and other autoimmune diseases" is very interesting and authors have answered all the comments by the reviewer. Therefore, I suggest to accept the manuscript without any delay.